# Overcoming False Illusions in Real-World Face Restoration with Multi-Modal Guided Diffusion Model

**Keda Tao[1], Jinjin Gu[2]\*, Yulun Zhang[3], Xiucheng Wang[1], Nan Cheng[1,4]\***

[1]Xidian University, [2]The University of Sydney, [3]Shanghai Jiao Tong University
[4]State Key Laboratory of Integrated Services Networks
KD.TAO@outlook.com, jinjin.gu@sydney.edu.au, yulzhang@sjtu.edu.cn
xcwang_1@stu.xidian.edu.cn, dr.nan.cheng@ieee.org

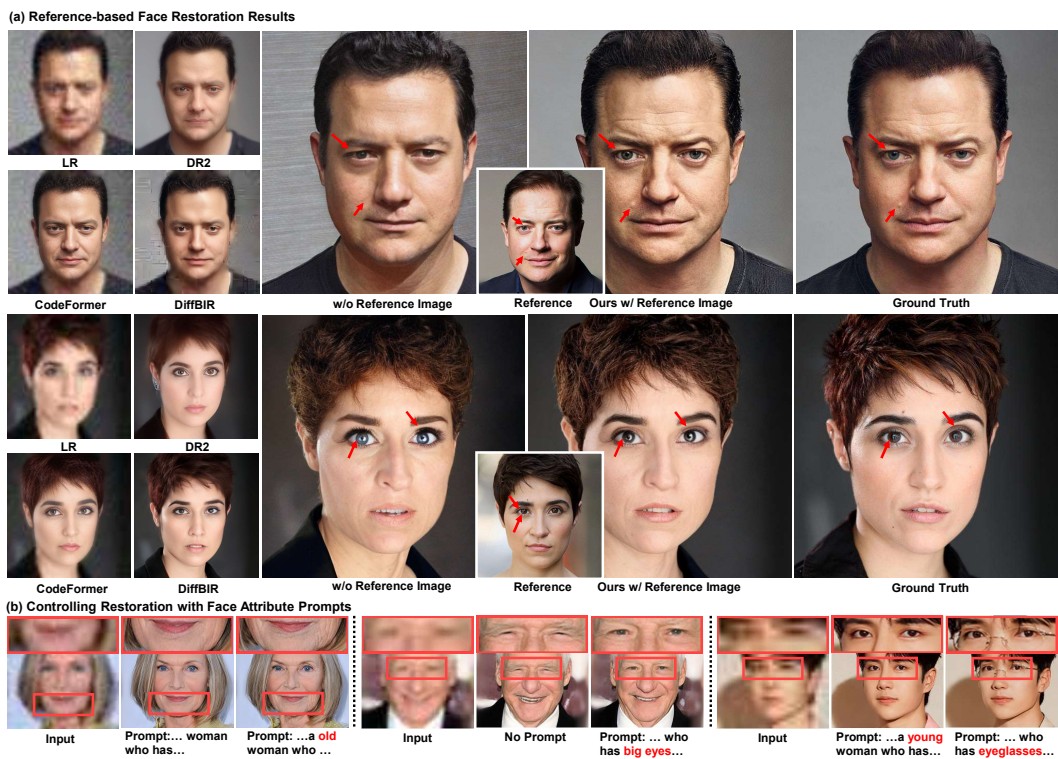

Figure 1: The proposed MGFR model demonstrates an exceptional ability in restoring low-quality face images, yielding more outstanding visual effects with the addition of reference images, particularly in situations of extreme degradation, shown in **(a)**. Furthermore, the model is capable of target-specific restoration in **(b)**, directed by facial attribute prompts. This encompasses defining facial age characteristics (Case 1), adjusting the restoration process based on attribute prompts (Case 2), and executing precise modifications to facial elements (Case 3). *w/o Reference Image* means the results of our model without introducing reference image.

## Abstract

We introduce a novel Multi-modal Guided Real-World Face Restoration (MGFR) technique designed to improve the quality of facial image restoration from low-quality inputs. Leveraging a blend of attribute text prompts, high-quality reference images, and identity information, MGFR can mitigate the generation of false facial attributes and identities often associated with generative face restoration methods. By incorporating a dual-control adapter and a two-stage training strategy, our method effectively utilizes multi-modal prior information for targeted restoration tasks. We also present the Reface-HQ dataset, comprising over 21,000 high-resolution facial images across 4800 identities, to address the need for reference face training images. Our approach achieves superior visual quality in restoring facial details under severe degradation and allows for controlled restoration pro-

---

*Corresponding authors

cesses, enhancing the accuracy of identity preservation and attribute correction. Including negative quality samples and attribute prompts in the training further refines the model's ability to generate detailed and perceptually accurate images.

# 1 INTRODUCTION

Real-World Face Restoration (FR) aims to reconstruct high-resolution, high-quality (HQ) facial images from their degraded, low-resolution observations. Recent works, leveraging powerful generative priors and diffusion models, have achieved significant progress (Menon et al., 2020; Yang et al., 2021b; Wang et al., 2021b; Lin et al., 2023; Wang et al., 2023b), particularly in addressing severely degraded facial images. However, the information contained in the low-quality (LQ) inputs is limited. FR inevitably introduces the illusion of generation, producing results with different facial attributes or even different identities from the target image. For example, in Figure 1 (a) and Figure 2, we cannot effectively predict the eye colour and skin characteristics of the person in the LQ input, resulting in the output results – even the quality can be improved – having an apparent perceptual distance from the target image. Many applications find this unacceptable, as humans can readily identify these flaws. Achieving optimal facial image recovery requires effectively tackling false hallucinations.

Practically, we find that for the restoration of specific face images, we can obtain a lot of prior information. For example, we may know this person's various attributes and identity, and there may even be other clear images of this person in the photo album. Suppose we can use this information as additional guidance to guide the restoration. In that case, we can alleviate the impact of false illusions on key issues, thus helping to generate facial details that better suit our needs. For example, in Figure 2 (a), when we provide an additional key description of gender and age, we can correct the illusion. In Figure 1 (a) and Figure 2 (b), additional high-quality images are used as reference, and the details of the eyes and skin texture can be accurately generated. What is even more gratifying is that this kind of prior information can be widely obtained, making this problem of application significant.

This work proposes a method called Multi-modal Guided Real-World Face Restoration (MGFR). We aim to use multiple control methods to consider diverse multi-modal prior information in FR to restore face images in a targeted manner. Specifically, MGFR uses attribute text prompts, HQ reference images, and identity information as priors for collaborative guidance during restoration. We designed a dual-control adapter with a two-stage training strategy to balance the complex multi-modal and multi-source prior information. This dual controller is compatible with pre-trained generative diffusion models (Rombach et al., 2022) and prioritizes restoration tasks while incorporating additional multi-modal guidance. In addition, we collect the **Reface-HQ** dataset to address the scarcity of reference image samples containing over 4800 identities and 21000 high-resolution facial images. Based on the FFHQ (Karras et al., 2019a) and the proposed Reface-HQ datasets, we develop a high-quality synthetic dataset for model training enriched with attribute text prompts. Furthermore, we adopt a counterintuitive strategy to integrate negative-quality samples with negative-quality prompts and negative-attribute prompts into training to enhance perceptual quality and detail generation.

The proposed MGFR model shows exemplary performance in the FR task, achieving superior visual quality in facial details, especially under severe degradation conditions. MGFR can take a high-resolution reference image as prior information and restore important details based on the reference image that cannot be displayed in the LQ input. The identity information provided by the reference image will also be considered in FR to ensure that the restoration does not change the identity characteristics. In addition, MGFR can also provide a certain degree of control over the restoration process through attribute text prompts, significantly enhancing the feasibility of FR. As shown in Figure 1 (b), textual prompts fulfil a dual function: they significantly reduce facial attribute illusions, such as "big eyes" or "old", and also guide the restoration of specific facial features, such as "wearing glasses" and "young".

# 2 RELATED WORKS

**Real-World Face Image Restoration**    Real-world face restoration (FR) concentrates on the challenging task of reconstructing HQ face images from LQ inputs. These LQ inputs are often blemished by various forms of quality degradation, such as low-resolution (Chen et al., 2018; Dong et al., 2014; Lim et al., 2017), blur (Kupyn et al., 2018; Shen et al., 2018), noise (Zhang et al., 2017),

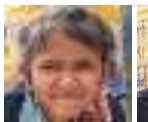 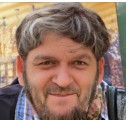 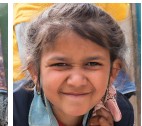 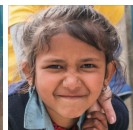 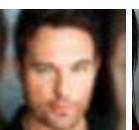 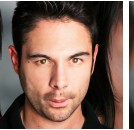 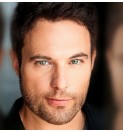 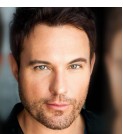

**(a)**     LQ      No Prompt     *'woman, young, no beard...'*     GT     **(b)**    LQ    w/o Reference    w/ Reference    GT

Figure 2: **Motivation.** In conditions of severe degradation, the loss of facial identity information becomes profoundly pronounced without a reference image. During the face restoration process, distortions of facial attributes, including gender and age, are commonly encountered. Appropriate attribute prompts can offer additional reference points and exert control in the recovery process.

and JPEG compression artifacts (Dong et al., 2015), *etc*. FR heavily relies on facial priors, such as facial landmarks (Chen et al., 2018), parsing maps (Chen et al., 2018; 2021), and facial component heatmaps (Yu et al., 2018). Generative priors (Karras et al., 2020; Rombach et al., 2022; Gu et al., 2020; Shen et al., 2020) have also emerged as fundamental elements in providing vibrant textures and details in FR (Menon et al., 2020; Hu et al., 2023; Zhu et al., 2022). Advanced techniques like GPEN (Yang et al., 2021a), GFP-GAN (Wang et al., 2021a), and GLEAN (Chan et al., 2021) are recognized for more effectively incorporating these priors within encoder-decoder structures. There are also works that considerably reduce the uncertainty commonly associated with generative priors (Gu et al., 2022; Zhou et al., 2022; Wang et al., 2022), which are trained on discrete feature codebooks for high-quality facial images. Recently, diffusion models like DiffBIR (Lin et al., 2023) have revitalized interest in this area, leveraging the generative power of pre-trained LDM as a prior. DR2 (Wang et al., 2023b) also contributes by transforming input images into noisy states and then denoising them to capture the essential semantic information.

**Reference-Based Face Image Restoration**    Reference-based face restoration utilizes HQ images of the same identity as references. This concept was first introduced in (Li et al., 2018a). To address discrepancies in poses and expressions, GWAInet (Dogan et al., 2019) and the later work of Li et al. (Li et al., 2020b; 2018b) focused on more effectively directing deformations or choosing the optimal reference image for reconstruction. MyStyle (Nitzan et al., 2022) adopts a unique approach by refining StyleGAN (Karras et al., 2019a) with numerous reference images based on personal appearance. DMDNet (Li et al., 2023) employs a dictionary constructed from diverse, high-quality facial images to rehabilitate degraded images using its high-quality components. In the MGFR framework, incorporating a single reference image is vital for tailoring the restoration to individual faces. Unlike conventional methods, MGFR does not require strict alignment constraints on expressions or postures.

**Multi-modal Guided Generation**    Diffusion models have shown significant effectiveness in a broad range of image processing tasks. Current methods (Chen et al., 2023b; Zhang et al., 2023; Yu et al., 2024; Chen et al., 2023a) employ pre-trained text-to-image diffusion models (Rombach et al., 2022) for image processing, demonstrating the potential of language as a comprehensive input for image reconstruction tasks. Concurrently, approaches like ControlNet (Zhang et al., 2023), T2I-adapter (Mou et al., 2023), and ControlNet-XS (Zavadski et al., 2023) have further developed the integration of more intricate condition controls within the text-to-image framework, facilitating more precise and tailored image generation. Nevertheless, the field of FR, particularly in the utilization of natural language prompts, continues to be an area of untapped potential.

## 3   METHODOLOGY

The proposed MGFR method is able to take face attribute text prompts, reference images, and identity information as input to alleviate illusions and improve visual effects. MGFR involves controlling information from multiple modalities. However, we found that if the model is directly trained to process control information from multiple sources and of different importance, it is not easy to utilize all the information effectively. The model may ignore the more complex information to utilize. This causes some of the controls to fail and reduces image quality. In our method, text prompts are the most complex control information because they involve understanding text and the correspondence between text and face attributes. Therefore, we divide the training into two stages. In the first stage, the training focuses on the basic text-guided restoration model (Section 3.1). This allows the model to restore high-quality images and understand facial attributes. Then, we introduce other control information on this basis. The second stage introduces the HQ reference image and face identity

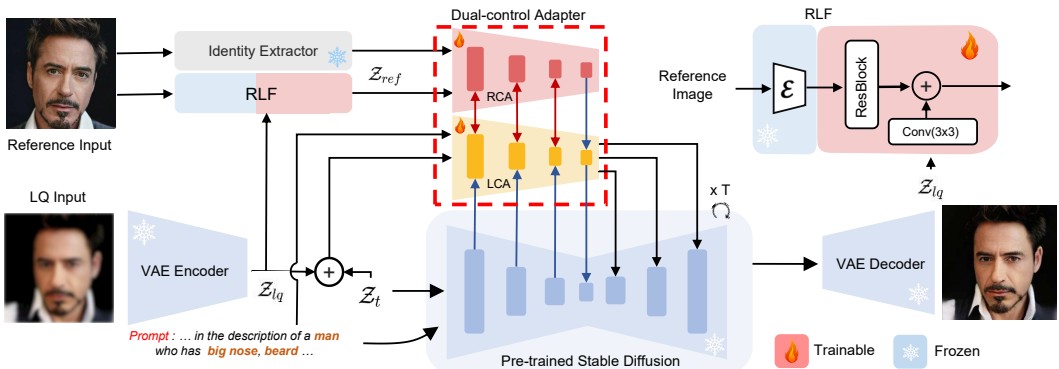

Figure 3: **Framework Overview.** This figure illustrates the overall workflow of the proposed MGFR model.

information as the control means (Section 3.2). To improve the image effect further, Section 3.3 describes negative examples and the adopted prompting strategy.

### 3.1 STAGE ONE: TEXT-GUIDED FACE RESTORATION

This stage trains the face image restoration model that accepts text prompt as input, as shown in Figure 4. We use the pre-trained Stable Diffusion (SD) (Rombach et al., 2022) model as our generative prior and train an additional adapter to extend it to the face restoration applications. The pre-trained SD generative prior has the ability to understand face image attributes and text and generate high-quality face images. In this stage, our model restores the image $x$ according to the condition $\{y, c_a\}$, where $y$ represents the degraded LQ image, and $c_a$ constitutes the facial attribute prompts describing the face attributes. We first use the CLIP text encoder (Radford et al., 2021) to calculate the text embedding $e_r = \mathrm{CLIP}(c_a)$. The LQ input $y$ is also mapped to a latent representation $z_{lq}$ using the VAE encoder in SD (Rombach et al., 2022). We then perform diffusion generation on this latent representation. In the framework of SD, the model uses UNet (Ronneberger et al., 2015) denoising model $\mathcal{E}_\theta (z_t, t, e_r)$ to perform the diffusion generation process, where $t$ is the time stamp in diffusion model and $z_t$ is the intermediate results at time $t$. Based on the ControlNet (Zhang et al., 2023) framework, we use an external adapter that takes the LQ input $y$ and text prompts embedding $e_r$ as input to provide guidance for the fixed UNet $\mathcal{E}_\theta$. We call this adapter the LQ Control Adapter (LCA). Specifically, the UNet model contains the encoder, intermediate blocks, and the decoder. The decoder receives features from the encoder and fuses them at each corresponding scale. The LCA contains the same encoder and intermediate blocks as the UNet model. The feature output of each scale in LCA is integrated with the corresponding scale of the UNet decoder to achieve the effect of output control. However, we found that simply using the above ControlNet framework has a key limitation – the lack of information exchange from the UNet encoder to the LCA. This gap means that the LCA is unaware of the processes that are performed in the UNet encoder, thereby limiting its ability to generate effective control features. In order to solve this problem, we add the feature output of each scale in the UNet encoder to the corresponding scale in the LCA. The LQ controller part of Figure 5 illustrates this operation. In this way, the capability of the LCA is greatly enhanced, so better visual effects and control results can be achieved.

### 3.2 STAGE TWO: MULTI-MODAL-GUIDED FACE RESTORATION

After the first stage of training, the model can already reconstruct high-quality images from the LQ inputs guided by text prompts. Next, we further enrich the guidance and introduce high-quality reference images and face identity information as additional control means based on the first-stage model. We design a new Dual-Control Adapter (DCA), as shown in Figure 3. In DCA, we introduce a Reference Control Adapter (RCA) specifically for reference image processing. RCA has the same architecture as LCA, and its role is to extract related and useful information from reference images and identity information and provide additional details to LCA. The input of RCA includes an HQ reference face image $r$ containing the same identity as the LQ input and its identity information embedding $e_f$. For the reference image $r$, we first use the VAE encoder consistent with the SD model for feature extraction to obtain $z_{ref}$. Next, we fuse the LQ latent representation $z_{lq}$ with $z_{ref}$ using a reference and LQ feature fusion module (RLF). This module allows RCA to identify

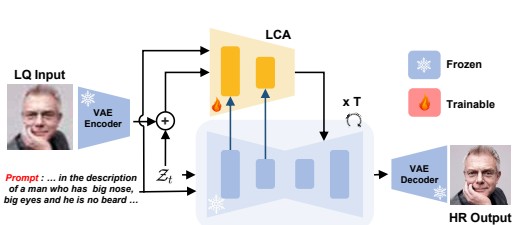

Figure 4: The model architecture employed during the initial training stage is discussed. In the article, 'Ours w/o Reference Image' refers to the outcome of the model trained following this stage.

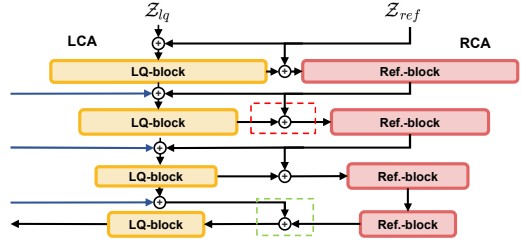

Figure 5: **Dual-control Adapter.** LQ-blocks are from the LQ control adapter (LCA), and Ref.-blocks are from the reference control adapter (RCA). $\oplus$ represents the element-wise add operation.

the high-frequency details missing in the LQ input and perform targeted information extraction for restoration guidance. For identity information embedding, we calculate $e_f = \text{Proj}(\text{Arcface}(r))$, where $\text{Arcface}(r)$ is the face recognition model Arcface (Deng et al., 2019) to extract the identity feature from the reference image $r$. We align it to the space that RCA can handle through a trainable linear projection layer $\text{Proj}(\cdot)$. Due to the function of the RCA extracting information from the reference image according to the LQ input, the RCA requires the information of the LCA branch as input. At the same time, RCA needs to provide the extracted information back to LCA in reverse. Therefore, we designed a dual-way interaction mechanism for RCA and LCA, as shown in Figure 5. In this design, RCA runs in parallel with LCA. At each scale, the LQ block in LCA first processes the fused information of both two branches and then hands the intermediate features to RCA. RCA performs feature extraction and processing based on these intermediate results and reference conditions and finally uses the same operation to apply the processing results to the next layer of LCA processing. Finally, the output of each scale of LCA is applied to the corresponding position of the UNet decoder. RCA directly affects LCA and, therefore, also affects the calculation of UNet. Since then, we have had a dual-control adapter that can accept multiple control inputs.

### 3.3 NEGATIVE SAMPLES AND PROMPT

Classifier-Free Guidance (CFG) (Ho & Salimans, 2022) introduces a novel control mechanism utilizing negative prompts to delineate unwanted content for the model. This feature can be leveraged to inhibit the generation of low-quality images by the model and to enhance the precision of facial detail reconstruction. Throughout the inference phase, at each step of diffusion, three distinct predictions are generated: one employing the positive prompt $pos$, another using the negative quality prompt $nq$, and a third via the negative attribute prompt $na$ (the negation sentence described by $pos$). We combine the results generated from these different prompts to form the final output:

$$\tilde{z}_{t-1} = z_{t-1}^{pos} + \lambda_{nq} \times (z_{t-1}^{pos} - z_{t-1}^{nq}) + \lambda_{na} \times (z_{t-1}^{pos} - z_{t-1}^{na}), \tag{1}$$

where $\lambda_{na}$ and $\lambda_{nd}$ is the hyperparameters, $z_{t-1}^{pos} = \mathcal{E}_\theta(z_t, z_{lq}, z_{ref}, t, pos)$, $z_{t-1}^{nq} = \mathcal{E}_\theta(z_t, z_{lq}, z_{ref}, t, nq)$, $z_{t-1}^{na} = \mathcal{E}_\theta(z_t, z_{lq}, z_{ref}, t, na)$. $pos$ represents a standard description of a facial attribute. $nq$ is the negative words of quality, e.g., "*oil painting, cartoon, blur, dirty, messy, low quality, deformation, low resolution, over-smooth*". $na$ is used for a negative description of a facial attribute, implying complete negation. Accurate prediction in both positive and negative directions is essential for the CFG technique. The lack of negative-quality samples and prompts in our training might cause the model to misinterpret negative prompts, leading to artifacts. To resolve this, we generated 16K images with negative-quality prompts using the original SD generative model and included these low-quality images in our training to enable the model to learn the concept of negative quality. Figure 9 (a) shows an example of the negative quality sample and prompt.

## 4 EXPERIMENTS

### 4.1 EXPERIMENTAL SETTING

**Datasets.** Our two-stage training method requires different datasets for training. For the first stage, we mainly train the model's ability to restore HQ images and process text prompts. Therefore, we need HQ images with text annotations for training. We synthesize training image pairs using the FFHQ dataset (Karras et al., 2019b). FFHQ contains 70,000 high-resolution face images, and we resize these images to $512 \times 512$ for training. In the second stage, in addition to requiring HQ face images to create training image pairs, we also need to assign HQ reference images with consistent identities but different details to each image. Although there are some datasets proposed

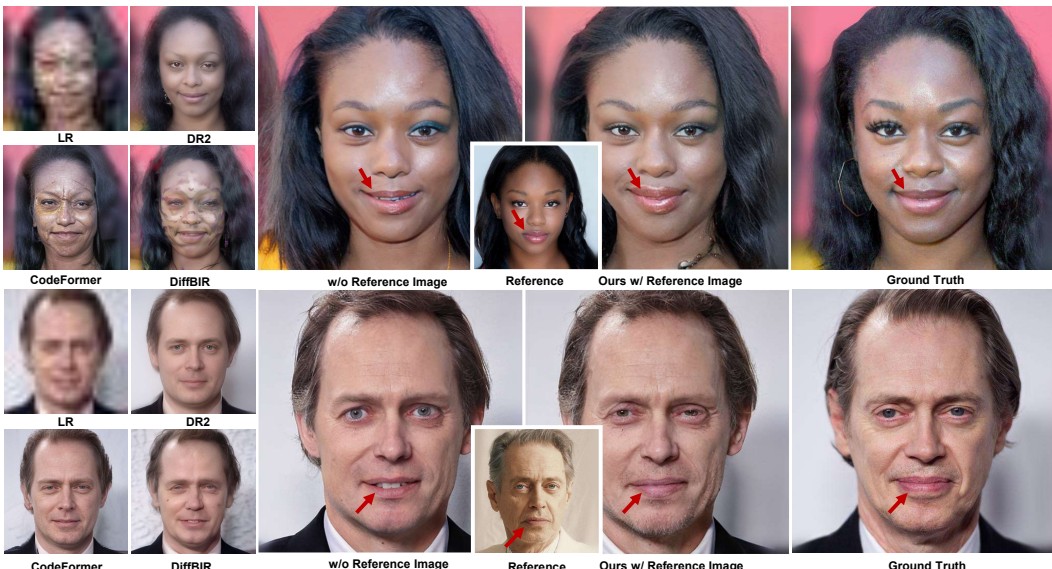

Figure 6: The MGFR model demonstrates a remarkable capacity for restoring LQ images. Upon integrating the reference image, particularly in instances of severe degradation, the model significantly enhances the restoration of facial details and overall image quality.

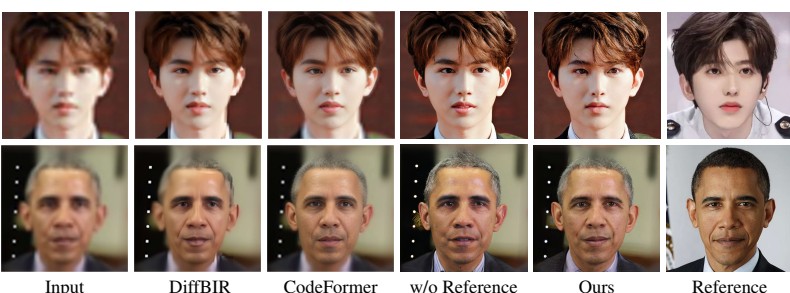

Figure 7: In the qualitative comparison of real-world low-quality (LQ) images, MGFR demonstrates success in recovering facial details without false illusion and preserving identity from. Please zoom in for a better view.

for reference face restoration (Liu et al., 2015; Yi et al., 2014), the resolution and quality of these datasets cannot meet the current requirements. In this work, we collect a new dataset for referenced face restoration called Reface-HQ. Reface-HQ contains 21,500 high-quality and diverse images of over 4800 identities. Additional details of Reface-HQ can be found in the Appendix A. To synthesize LQ images, we follow the degradation model and setting used in (Wang et al., 2023b). Our test data also involves multiple sources, including CelebA-Test (Liu et al., 2015), Reface-Test, and real-world LQ images collected from the Internet. Specifically, CelebA-Test contains 3,000 testing images from the CelebA-HQ dataset. Reface-Test contains 1,300 images of 280 identities split from the proposed Reface-HQ dataset. The LQ images for testing are synthesized within the same degradation range as the training setting.

**Attribute Prompt.** Text prompts are important for us to control face attributes and improve quality. In our method, a total of three types of prompts are introduced. Two attribute prompts describe the face attributes, and the last one describes the negative quality of the image. For attribute prompts, $pos$ contains positive descriptions of face attributes, while $na$ describes attributes that do not exist in this image to provide negative prompts of attributes. To obtain these descriptions, we first use a pre-trained face attribute detector (He et al., 2017) to extract the presence of each attribute in the face. We considered 28 different attributes in this work. For attributes that have high confidence to exist, we add them to the $pos$ positive attributes. For the remaining attributes with low confidence, we classify

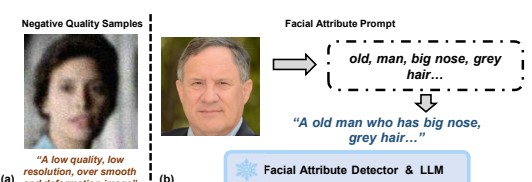

Figure 9: **Training Data Composition.** Initially, negative quality samples are incorporated into the training to enhance the clarity and quality of the restored image. Furthermore, large language models, coupled with a facial attribute classifier, are employed to extract attribute texts for integration into the training.

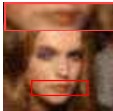 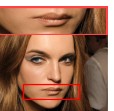 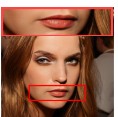 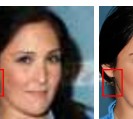 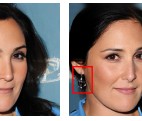 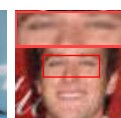 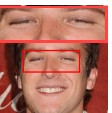 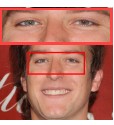

| Input | No Prompt | Add: *'lipstick'* | Input | No Prompt | Add: *'earrings'* | Input | No Prompt | Add: *'big eyes'* |

Figure 8: MGFR demonstrates capability of face image restoration facilitated by text prompts. It possesses the capacity to artificially modulate specific aspects of the restoration outcomes, such as determining the presence of accessories like lipstick or glasses (Cases 1 & 2), and orchestrating the restoration process in alignment with facial attributes (Case 3).

| Method | Real-SR(×4) | | | | Real-SR(×8) | | | | Real-SR(×16) | | | |
|---|---|---|---|---|---|---|---|---|---|---|---|---|
| | LPIPS ↓ | ManIQA | ClipIQA | MUSIQ | LPIPS ↓ | ManIQA | ClipIQA | MUSIQ | LPIPS ↓ | ManIQA | ClipIQA | MUSIQ |
| PSFRGAN | 0.2938 | 0.5927 | 0.5702 | 73.39 | 0.3315 | 0.6015 | 0.5956 | 73.08 | 0.3788 | 0.5739 | 0.6274 | 71.76 |
| GPEN | 0.2828 | 0.6596 | 0.6430 | 69.25 | 0.3217 | 0.6754 | 0.6299 | 68.63 | 0.3831 | 0.6618 | 0.5897 | 66.61 |
| VQFR | 0.2951 | 0.2875 | 0.2490 | 62.95 | 0.3277 | 0.4163 | 0.2363 | 61.92 | 0.3761 | 0.6513 | 0.2148 | 60.49 |
| CodeFormer | 0.2927 | 0.5803 | 0.5179 | 75.47 | 0.3193 | 0.5970 | 0.6235 | 75.09 | 0.3821 | 0.5803 | 0.5877 | 70.85 |
| DR2 | 0.3264 | 0.5749 | 0.4441 | 63.43 | 0.3580 | 0.5246 | 0.4494 | 59.46 | 0.3796 | 0.5160 | 0.5035 | 70.31 |
| DiffBIR | 0.2611 | 0.6068 | 0.7681 | 74.27 | 0.3017 | 0.6058 | 0.7439 | 73.87 | 0.4238 | 0.5361 | 0.7164 | 67.41 |
| BFRffusion | 0.3258 | 0.5477 | 0.5572 | 45.32 | 0.3739 | 0.4404 | 0.5298 | 42.84 | 0.3735 | 0.4204 | 0.5098 | 43.16 |
| Ours w/o Reference | 0.2925 | 0.6854 | 0.8244 | 76.22 | 0.3227 | 0.6776 | 0.8083 | 75.94 | 0.3760 | 0.6729 | 0.7944 | 75.76 |

Table 1: **Quantitative Comparison in CelebA-Test.** Results in red and blue signify the highest and second highest, respectively. The ↓ indicates metrics whereby lower values constitute improved outcomes, with higher values preferred for all other metrics.

them as $na$ negative attributes. At this time, these attributes are still separate words. We use a large language model to organize the separated words into natural language to facilitate the understanding of the CLIP text encoder. Thus, each face image is associated with two attribute prompts detailing existing and non-existing attributes. For the negative quality prompt, $nq$ involves "low quality, low resolution, over-smoothed and distorted images", as shown in Figure 9 (a). See Appendix B for comprehensive details on the training and inference procedures involving attribute prompts.

**Implementation.** The training involved fine-tuning based on Stable Diffusion v2.1 (Rombach et al., 2022), with the control adapter structure adhering to (Zhang et al., 2023). The Adam optimizer (Kingma & Ba, 2014) was employed, featuring a learning rate of $e^{-5}$. The initial training stage spanned 15 days, while the subsequent stage lasted 5 days, utilizing 4 Nvidia A100 GPUs with a batch size of 4. For testing purposes, the hyperparameters were set as $T = 500$, $\lambda_{na} = 0.5$ and $\lambda_{nq} = 0.5$.

**Metrics.** For quantitative comparison, followed by many previous works (Lin et al., 2023; Yu et al., 2024), the selected metrics include full-reference metrics PSNR, SSIM, and LPIPS (Zhang et al., 2018), as well as non-reference metrics ManIQA (Yang et al., 2022), ClipIQA (Wang et al., 2023a), and MUSIQ (Ke et al., 2021). Furthermore, the Arcface identity distance (Deng et al., 2019) (ID) is utilized to assess the similarity of identity information.

## 4.2 COMPARISONS WITH STATE-OF-THE-ART METHODS

MGFR is qualitatively and quantitatively compared with state-of-the-art methods in FR. Notably, the model trained in the initial stage, which is a restoration model solely guided by attribute prompts, already achieves superior visual results. The non-reference prior-based methods selected include PSFRGAN (Chen et al., 2021), GPEN (Yang et al., 2021a), VQFR (Gu et al., 2022), CodeFormer (Zhou et al., 2022), DR2 (Wang et al., 2023b), BFRffusion (Chen et al., 2024) and DiffBIR (Lin et al., 2023), along with reference prior-based methods ASFFNet (Li et al., 2020b) and DMDNet (Li et al., 2023). Particularly, to ensure contrastive fairness during the inference stage, the description text, containing restricted attributes, is obtained through low-resolution processing. In practical applications, however, users can freely set attribute prompts, enabling more precise and comprehensive guidance. For qualitative results comparing ASFFNet and DMDNet, please refer to the Appendix.

**Comparison on Synthetic Degradations.** Firstly, a quantitative comparison of our model without reference images on the synthetically degraded CelebA-Test dataset is conducted without reference image guidance. According to Table 1, our model achieves the best results on all non-reference metrics, indicative of the superior image quality of the results. Due to space limitation, the values of SSIM and PSNR of Table 1 are shown in Appendix C.1. Additionally, the method's limitations on full-reference metrics are also noted. This phenomenon, preliminarily demonstrated by experiments

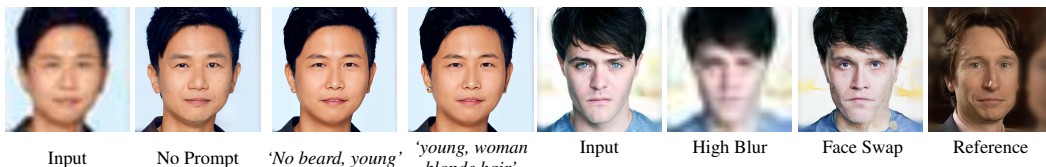

Input | No Prompt | *'No beard, young'* | *'young, woman blonde hair'* | Input | High Blur | Face Swap | Reference

Figure 10: Attribute prompts that manifestly contravene low-resolution inputs prove ineffectual and result in distortions and artifacts within the restored image.

Figure 11: Face Swapping: MGFR is capable of leveraging the reference map to alter the comprehensive components of the face.

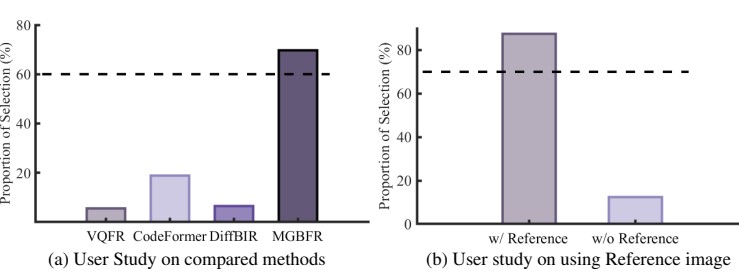

(a) User Study on compared methods

(b) User study on using Reference image

Figure 12: The results of our user study. We randomly select face images under multiple test datasets for user study. Our model achieves excellent recovery quality, which can be further enhanced with high-quality reference image and identity information guidance.

in (Yu et al., 2024; Jinjin et al., 2020), necessitates a reevaluation of the reference value of indicators like PSNR, SSIM, LPIPS, and the proposal of more effective methods to assess advanced FR methods, particularly as quality improves. More qualitative comparison results of our model can be found in Appendix C. Subsequently, Figure 6 and Figure 1 (a) present a qualitative comparison of the MGFR method applied to the Reface-Test dataset. Even in cases of severe degradation, our method successfully produces highly superior facial details guided by the reference image. In addition, we provide a comparison between our model without reference images and MGFR, with a particular focus on FR tasks involving features like double eyelids, pupil color, and finer facial details, such as wrinkles and moles, which cannot be accurately captured without reference image guidance. This further demonstrates the superiority of utilizing reference image guidance in the FR task. Finally, Table 2 offers quantitative comparison results, indicating that our method significantly surpasses other state-of-the-art methods in perceived quality.

We also conducted a user study with a total of 40 participants, comparing MGFR to other approaches. Participants were asked to select the best quality recovery result from these test techniques for each pair of comparison images, or if no reference image was provided, the result that came closest to the Ground Truth. Section 4.2 presents the results, which demonstrate that our method outperforms the state-of-the-art methods in terms of recovery quality. Furthermore, the reconstruction effect can be further enhanced by using the reference image guidance.

**Comparison on Real-world Degradations.** Additionally, our method was tested on real-world LQ images, which involved collecting degraded face images of publicly available images alongside reference images. The qualitative results, presented in Figure 12, demonstrate that the resulting images possess realistic visual effects with minimal facial illusions. More results are presented in Appendix C.

| Degradation | Method | PSNR | SSIM | LPIPS ↓ | ManIQA | ClipIQA | MUSIQ | ID ↓ |
|---|---|---|---|---|---|---|---|---|
| ×8 | ASFFNet | 23.43 | 0.6811 | 0.2452 | 0.5685 | 0.6215 | 71.66 | 0.7053 |
| | DMDNet | 23.85 | 0.7062 | 0.2667 | 0.5023 | 0.6023 | 72.31 | 0.6964 |
| | DR2 | 23.58 | 0.6581 | 0.2532 | 0.5340 | 0.5956 | 69.00 | 0.7957 |
| | CodeFormer | 23.88 | 0.6904 | 0.2912 | 0.4959 | 0.5823 | 74.80 | 0.6579 |
| | DiffBIR | 24.12 | 0.6717 | 0.2785 | 0.5547 | 0.7474 | 73.73 | 0.6379 |
| | MGBFR(Ours) | 23.10 | 0.6248 | 0.2688 | 0.6535 | 0.8147 | 75.51 | 0.5166 |
| ×16 | ASFFNet | 21.70 | 0.6472 | 0.3013 | 0.5803 | 0.6221 | 71.57 | 0.9361 |
| | DMDNet | 22.37 | 0.6761 | 0.3179 | 0.4579 | 0.4727 | 67.27 | 0.9270 |
| | DR2 | 22.28 | 0.6720 | 0.3269 | 0.5233 | 0.5693 | 66.39 | 0.8676 |
| | CodeFormer | 21.88 | 0.6124 | 0.3400 | 0.5547 | 0.5855 | 71.30 | 0.8658 |
| | DiffBIR | 21.51 | 0.5939 | 0.3944 | 0.4937 | 0.7144 | 67.42 | 0.8876 |
| | MGBFR(Ours) | 21.75 | 0.6033 | 0.2989 | 0.6524 | 0.8046 | 75.06 | 0.7401 |

Table 2: **Quantitative Comparison in Reface-Test.** Quantitative comparison of guided recovery results based on reference images. DR2, CodeFormer, and DiffBIR do not use reference images.

### 4.3 CONTROLLING RESTORATION WITH ATTRIBUTES PROMPTS

Our method facilitates targeted image restoration guided by attribute prompts. As illustrated in Figure 8, the comparison between the first and second cases reveals that the integration of supplementary attribute prompts facilitates the manipulation of subtle facial attributes absent in the original image. This includes the addition of glasses, earrings, and accessories. In scenarios of severe degradation,

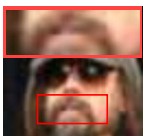 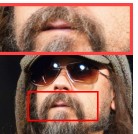 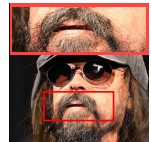 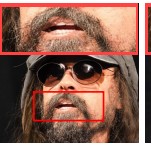 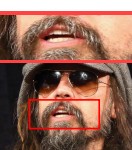

Figure 13: Negative quality prompts engender restoration outcomes characterized by high definition, whereas negative attribute prompts yield results with enhanced detail.

| | | Use Negative | Use Negative | |
| Input | w/o Negative Prompt | Quality Prompt | Attribute Prompt | GT |

| Real-SR(×8) | SSIM | PSNR | LPIPS | ManIQA | ClipIQA | MUSIQ |
|---|---|---|---|---|---|---|
| w/o Link-UL | 0.6372 | 22.17 | 0.3275 | 0.5931 | 0.7082 | 71.33 |
| w/o Link-LR | 0.6659 | 23.86 | 0.2873 | 0.6152 | 0.6645 | 65.70 |
| MGBFR(Ours) | 0.6248 | 23.10 | 0.2688 | 0.6535 | 0.8147 | 75.51 |

Table 3: Ablation study of additional information exchange in the MGBFR model. 'w/o Link-LR' means that the upward flow of information from LCA to RCA is removed.

| Prompts | | | LPIPS ↓ | SSIM | PSNR | ManIQA | ClipIQA | MUSIQ |
|---|---|---|---|---|---|---|---|---|
| pos | nq | na | | | | | | |
| | | | 0.3264 | 0.6858 | 25.15 | 0.4782 | 0.2568 | 49.97 |
| ✓ | | | 0.2690 | 0.6484 | 24.43 | 0.6441 | 0.7008 | 73.26 |
| ✓ | ✓ | | 0.2930 | 0.6066 | 23.27 | 0.6656 | 0.7999 | 75.34 |
| ✓ | | ✓ | 0.2702 | 0.6511 | 24.49 | 0.6437 | 0.7029 | 73.11 |
| ✓ | ✓ | ✓ | 0.3227 | 0.5904 | 22.34 | 0.6776 | 0.8083 | 75.94 |

Table 4: Ablation study of attribute prompts and negative prompts

exemplified by the third case, reconstructing facial features like eyes poses a significant challenge without external prompts. More results are shown in Appendix E.1.

However, it is imperative to acknowledge that attribute prompts do not invariably yield efficacy. As demonstrated in Figure 10, our model is capable of control tuning through attribute prompts. However, prompts that starkly contradict LQ inputs, like "blonde hair", are found to be ineffective. This ensures the model's adherence to the provided LQ inputs. Furthermore, as illustrated by the input of an LQ male face in Figure 10, when the input attribute is "Female", the model subtly incorporates the attribute label "Female" into the image. This is achieved through modifications like the addition of an earring and the removal of the beard while remaining faithful to the LQ input. Such modifications further underscore the efficacy of attribute text in guiding the restoration process. This outcome is not unexpected. On the contrary, excessive control capability might lead to a reduction in the restoration effectiveness, countering the fundamental intent of image reconstruction efforts and thereby demonstrating the robustness of the proposed method.

## 4.4 ABLATION STUDY

**Attribute Prompt and Negative Samples.** Figure 13 displays qualitative results under various settings, aligning with the strategies outlined in Section 3.3. It can be seen that incorporating a negative quality prompt significantly enhances restoration quality, while the addition of a negative attribute prompt yields images with finer details. Quantitative results under various settings are also presented in Table 4. Adding either positive attribute prompts or negative quality prompts is observed to improve the perceived quality of the images significantly. Utilizing both types of prompts in conjunction with the negative attribute prompt achieves the most favourable perceived effect. The impact of hyperparameters on the results was also explored, revealing that settings of $\lambda_{na} = 0.5$ and $\lambda_{nq} = 0.5$ yield the best perceptual outcomes, balancing sharpness and definition. Please refer to Appendix G for detailed qualitative results in different hyperparameters.

**Face Swapping.** MGFR can facilitate face-swapping operations involving the processing of highly degraded LQ images to obscure identities, as shown in Figure 11. Face images and identity information from different identities are utilized as guides to achieve face swapping and identity replacement. Besides proposing an additional application for the model, this experiment further illustrates the method's efficacy in utilizing identity information and reference images for guidance.

**Additional Information Exchange.** Unlike (Zhang et al., 2023), we have integrated an additional information flow exchange link (Link-UL) from the U-net model to LCA, and a bidirectional information flow link (Link-LR) between LCA and RCA. Table 3 displays the quantitative test results for the presence of the aforementioned information exchange links. Notably, 'w/o Link-UL' refers to results obtained with a single information flow from LCA to U-net model. It is evident that additional information flow exchanges result in improved perceived quality.

**Arcface Identity Embedding.** Our model is able to leverage identity information to guide the image restoration process, aiming to mitigate the deficit in facial identity information substantially. As demonstrated in Figure 15, our model employs the identity encoding formulated by the identity information extractor to mitigate the deficiency of facial identity information in the restored image. After losing arcface identity embedding, the recovered results still have high quality but there is a false illusion of face identity information.

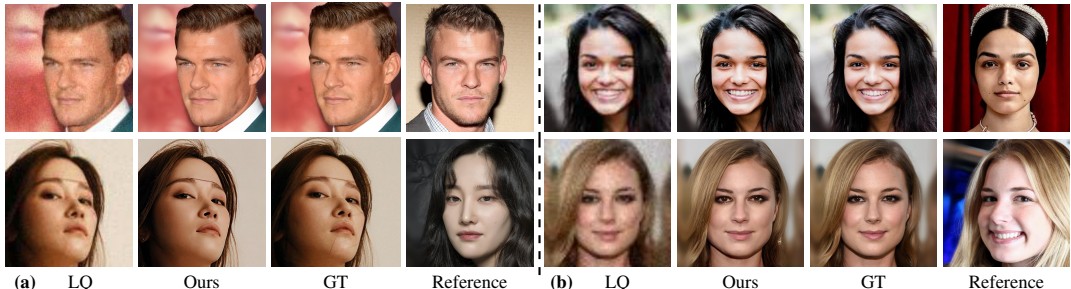

Figure 14: **Face restoration results with reference images with different expressions and poses.** When there are differences in pose and expression between the reference image and the low-quality input image, our model can still achieve a good restoration effect without the generation of artifacts.

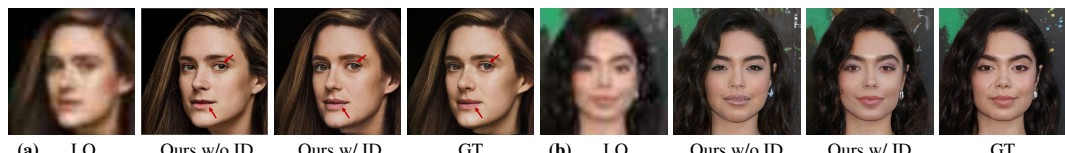

Figure 15: **Ablation experiments about arcface identity embedding (ID).** The additional identity embedding can greatly reduce the false illusion of identity information in the recovery results.

**Different expressions and poses reference images.** In previous studies, reference image-based face restoration has been widely explored, but its efficacy is constrained by the need for strict alignment between the reference image and the low-quality (LQ) input. As shown in Figure 21, the recovery results of ASFFNet and MDMNet exhibit severe distortions when the reference image and the LQ input are slightly misaligned. However, our method completely resolves this issue, as it imposes no strict requirements on the expression, pose, or other variations of the reference image. As demonstrated in Figure 14, even when there are discrepancies between the reference image and the LQ input, such as face orientation, labeling, makeup, or pose, our model consistently achieves high-quality restoration without any artifacts.

## 4.5 LIMITATIONS AND DISCUSSION

Although this represents an initial foray into attribute text-guided face image restoration, the flexibility of its text input is somewhat constrained due to the nature of the training samples. The model struggles to fully comprehend freely composed attribute description sentences, tending instead to rely on attribute labels embedded within fixed template text prompts, which limits its applicability in broader contexts. Furthermore, when users input attribute labels unseen during training, these do not effectively guide the recovery process. These limitations highlight the importance and necessity of utilizing high-quality data on a larger scale.

Moreover, we believe that excessive flexibility in controlling facial attributes poses a risk of model misuse. Our model demonstrates robust recovery even in the absence of attribute hints. Therefore, for the first time, we incorporate additional multimodal information to enhance recovery quality without increasing task complexity. The reference image serves solely to enhance facial details and does not alter the recovered results. In summary, we remain committed to the core objective of the recovery task: ensuring that the output remains faithful to the low-quality input. Furthermore, leveraging multiple reference images enables more precise detail recovery, while the exploration of utilizing multiple low-quality reference images for collaborative guidance is left for future research.

## 5 CONCLUSION

We introduce MGFR as a pioneering method in real-world face restoration at the cutting edge of face image restoration technology, capable of using multi-modal information for guidance to achieve realistic visual effects. Simultaneously, MGFR extends the possibilities of face restoration by controlling text prompts with attributes. The proposed Reface-HQ dataset also offers significant potential for advancing the development of face restoration models based on reference images. As the first multi-modal face image restoration model, MGFR establishes a new benchmark for future technological advancements.

ACKNOWLEDGE

This work was supported by the National Key Research and Development Program of China (2024YFB2907500).

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

APPENDIX

## A    REFACE-HQ DATASET

The bulk of prior reference-based face restoration methodologies commonly focus on training and testing with $256 \times 256$ images. This is primarily due to the limitations of existing datasets such as CelebA Liu et al. (2015), VggFace2 Cao et al. (2018), and CASIA-WebFace Yi et al. (2014), which offer reference images mainly for face or attribute recognition but do not include high-quality images suitable

| Dataset | Number of ID | Image | Size | Synthesized |
|---|---|---|---|---|
| CASIA-WebFace | 10575 | 494414 | 256×256 | ✗ |
| Celeba | 10,177 | 202599 | 178×218 | ✗ |
| IDiff-Face | - | - | 128×128 | ✓ |
| VggFace2 - HQ | 1200 | 24000 | 512×512 | ✗ |
| CelebRef-HQ | 1000 | 10000 | 512×512 | ✗ |
| Reface-HQ | 4800 | 21500 | 512×512 | ✗ |

Table 5: Datasets Comparison.

for training at higher resolutions, like $512 \times 512$ or $1024 \times 1024$, thereby limiting their practical applications. Additionally, high-definition datasets recently introduced, such as CelebRef-HQ Li et al. (2022) and VggFace2-HQ, face challenges in maximizing the potential of models due to their limited number of images and narrow range of identities.

To address this challenge, we have created a new **real-world** dataset named Reface-HQ, as shown in Figure 16. The Reface-HQ dataset encompasses high-definition facial images of celebrities, which have been collected from the Internet. Initially, images with inadequate resolution (minimum 512), low quality and outliers lacking facial features were eliminated. Subsequently, identities represented by fewer than two images were excluded, and face image crop alignment was conducted. Each identity was also manually inspected to eliminate discrepancies in age and makeup. Additionally, to enhance the fairness and inclusiveness of the algorithm, we meticulously review the dataset to ensure it includes samples from all races and skin colors. We strive to ensure the diversity of the training data, thereby minimizing algorithmic bias and discrimination, and further enhancing the algorithm's fairness and inclusiveness. In summary, Reface-HQ encompasses 4,800 identities, totaling 21500 images with a resolution of 512, subsequently partitioned into three segments: 4520 identities for the training set and 280 for the Reface-Test. The comparison of datasets available for special face restoration tasks is shown in Table 5. IDiff-Face Boutros et al. (2023) is a composite dataset with an indefinite number of images.

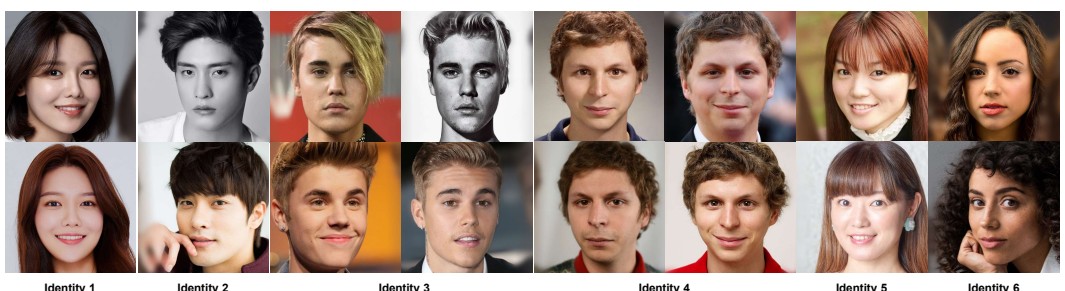

Figure 16: Demonstration of the Reface-HQ dataset.

### A.1    ABLATION EXPERIMENT

For diffusion models and adapter structures, both the quality and quantity of training data are critical factors affecting the model's final performance. Table 6 presents the quantitative comparison results of our proposed model under various training data volumes. It is evident that the model's performance significantly decreases with only 10,000 training data samples.

| Real-SR(×8) | SSIM | PSNR | LPIPS | ManIQA | ClipIQA | MUSIQ |
|---|---|---|---|---|---|---|
| 10K Training Samples | 0.6254 | 23.46 | 0.2535 | 0.6388 | 0.7424 | 72.23 |
| 20K Training Samples | 0.6248 | 23.10 | 0.2688 | 0.6535 | 0.8147 | 75.51 |

Table 6: Ablation Experiment about training.

# B    ATTRIBUTE PROMPT

This section provides a supplementary note on the attribute text prompts utilized in MGFR. For the training data, attribute labels are first extracted from the FFHQ or Reface-HQ dataset's face images using a facial attribute classifier. The 28 types of attributes included are listed in Table 7, while labels with binomial characteristics (such as Male and Female, no beard and beard,

| Row 1 | Row 2 | Row 3 | Row 4 |
|---|---|---|---|
| Black Hair | Blond Hair | Blurry | Brown Hair |
| - | Eyeglasses | Gray Hair | Heavy Makeup |
| Mouth Slightly Open | Mustache | Big Eyes | No Beard |
| Receding Hairline | Sideburns | Smiling | Straight Hair |
| Wearing Earrings | Wearing Hat | Male | Wearing Necklace |
| Big Nose | - | Wearing Lipstick | Young |
| Wavy Hair | Big Lips | Bald | Bangs |

Table 7: Face Attribute.

etc.) are not repetitively shown. Regarding the classification threshold, attributes with a probability greater than 0.6 are considered positive, those with a probability less than 0.4 as negative, and the rest as uncertain in describing facial features. LLM is utilized to embed the attribute labels into a descriptive sentence template, thereby enhancing the model's understanding. To augment the model's grasp of negative attribute descriptions, two sentences of prompt text are provided for each image, as illustrated in Figure 17. Both descriptions offer a positive portrayal of the face, with Prompt B specifically focusing on the negative attributes.

In the inference stage, following the approach detailed in Section 3.3, we apply positive attribute prompts ($pos$), negative quality prompts ($nq$), and negative attribute prompts ($na$) in each iteration. For example, in restoring a LQ image, if it is assumed to contain attributes like 'smiling, man, black hair, eyeglasses,' the corresponding text for image restoration can be generated as follows:

- **Positive Prompt:** A high quality, high resolution, realistic and extremely detailed image in the description of a smiling man who has black hair and eyeglasses.

- **Negative Attribute Prompt:** A high quality, high resolution, realistic and extremely detailed image not in the description of a smiling man who has black hair and eyeglasses.

- **Negative Quality Prompt:** A low quality, low resolution, over smooth and deformation image.

The underlying premise is to prevent our model from generating low-quality images and images with mismatched facial attributes. Extensive experiments demonstrate the effectiveness of our proposed attribute prompts.

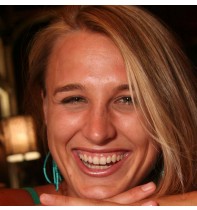

**Prompt A** : *A high quality, high resolution, realistic and extremely detailed image in the description of a smiling young woman who has big nose. she is wearing lipstick and she is no beard.*

**Prompt B** : *A high quality, high resolution, realistic and extremely detailed image not in the description of a old man who has bangs, big lips, black hair, blond hair, brown hair, eyeglasses, gray hair, straight hair, wavy hair.*

Figure 17: Attribute prompts composition in training.

## C   MORE QUALITATIVE COMPARISONS FOR OUR MODEL WITHOUT REFERENCE IMAGES.

This section presents qualitative comparisons experimental results of our model without reference images, focusing on attribute text-guided face recovery. Importantly, for a fair comparison, the attribute during the inference phase are derived from the LQ input, which means the model's maximum potential is not fully realized. We assert that in practical scenarios, users will be able to supply more precise attribute text for enhanced recovery guidance. Although, our model demonstrates the most superior visual effects and details when compared to other state-of-the-art methods.

| WebPhoto-Test | ManIQA | ClipIQA | MUSIQ |
|---|---|---|---|
| DR2 | 0.4868 | 0.6184 | 64.36 |
| DiffBIR | 0.4068 | 0.6858 | 55.73 |
| Ours | **0.5901** | **0.8397** | **72.52** |

Table 8: Quantitative comparison with other diffusion model-based methods on real-world degradations in WebPhoto-Test.

Figure 18 and Figure 19 display the qualitative comparison results of our model against other advanced models under conditions of mild and moderate degradation of LQ input, respectively. It is evident that the previous methods exhibit severe facial illusion, whereas our model attains the best visual outcomes. Notably, as shown in Figure 20 and Figure 21, our model demonstrates a remarkable ability to recover severely degraded input images with high quality and fidelity. Finally, Figure 22 shows the effect of restoration on **real-world** LQ inputs and Table 8 presents the quantitative comparison results between our model and the principal comparison methods using real-world LQ inputs from the WebPhoto-Test dataset.

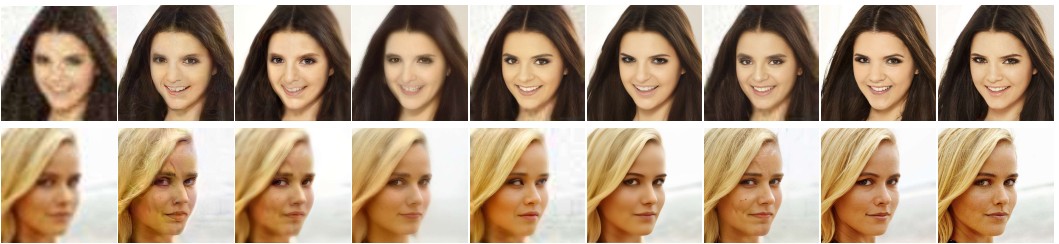

LR    PSFRGAN    GPEN    DR2    VQFR    DiffBIR    CodeFormer    Ours w/o Ref.    GT

Figure 18: More qualitative comparisons for our text-guided baseline model on synthetic dataset under mild degradation in CelebA-Test dataset. Zoom in for best view.

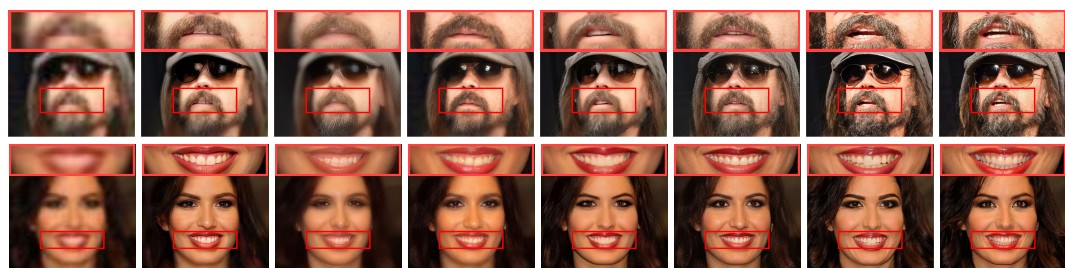

LR    GPEN    DR2    VQFR    DiffBIR    CodeFormer    Ours w/o Ref.    GT

Figure 19: More qualitative comparisons for our text-guided baseline model on synthetic dataset under moderate degradation in CelebA-Test dataset. Zoom in for best view.

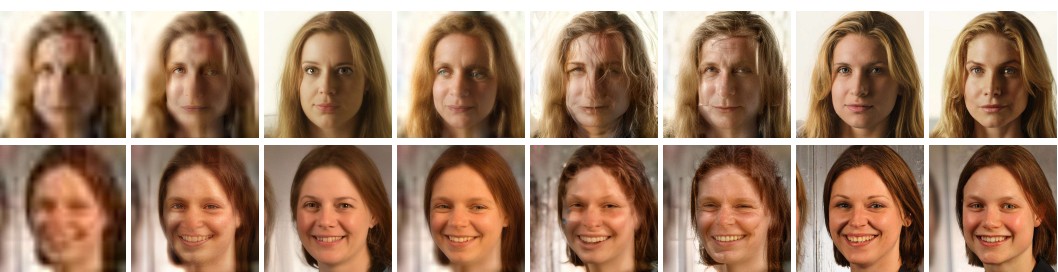

LR    GPEN    DR2    VQFR    DiffBIR    CodeFormer    Ours w/o Ref.    GT

Figure 20: More qualitative comparisons for our text-guided baseline model on synthetic dataset under severe degradation in CelebA-Test dataset. Zoom in for best view.

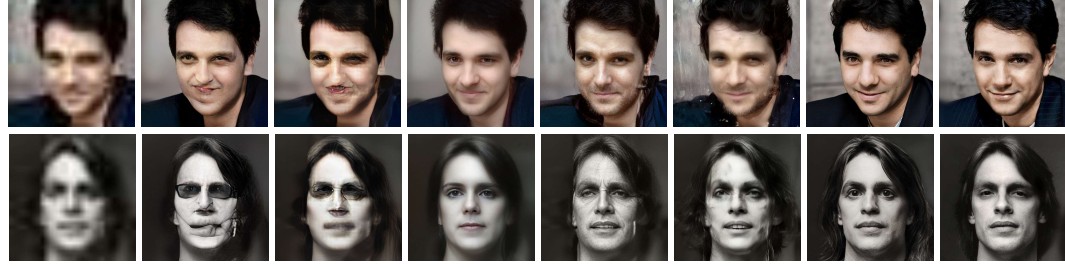

LR     ASFFNet     MDMNet     DR2     CodeFormer     DiffBIR     Ours w/o Ref.     GT

Figure 21: More qualitative comparisons for our text-guided baseline model on synthetic dataset under severe degradation in Reface-Test dataset. Zoom in for best view.

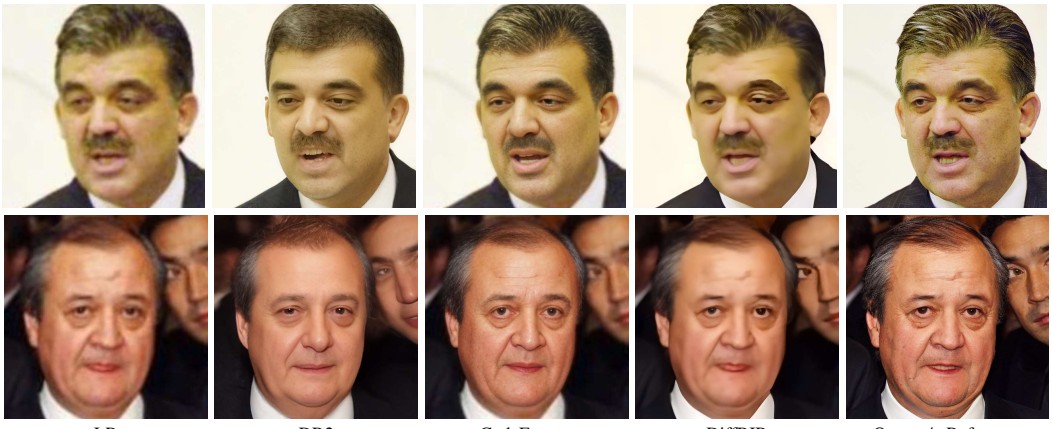

LR     DR2     CodeFormer     DiffBIR     Ours w/o Reference

Figure 22: More qualitative comparisons for our model without reference images on real world images. Zoom in for best view.

## C.1 ADDITIONAL SUPPLEMENT TO TABLE 1

Due to space limitation, for the quantitative comparison results of our model without reference images, Table 1 does not show the numerical values of PSNR and SSIM, and we supplement them in Table 9, Table 10 and Table 11.

| Method | Real-SR(×4) | | | | | |
|---|---|---|---|---|---|---|
| | LPIPS | PSNR | SSIM | ManIQA | ClipIQA | MUSIQ |
| PSFRGAN | 0.2938 | 23.72 | 0.6522 | 0.5927 | 0.5702 | 73.39 |
| GPEN | 0.2828 | 24.78 | 0.7056 | 0.6596 | 0.6430 | 69.25 |
| VQFR | 0.2951 | 23.81 | 0.6878 | 0.2875 | 0.2490 | 62.95 |
| CodeFormer | 0.2927 | 24.56 | 0.6809 | 0.5803 | 0.5179 | 75.47 |
| DR2 | 0.3264 | 23.74 | 0.6827 | 0.5749 | 0.4441 | 63.43 |
| DiffBIR | 0.2611 | 24.49 | 0.6778 | 0.6068 | 0.7681 | 74.27 |
| BFRffusion | 0.3258 | 24.87 | 0.7014 | 0.5477 | 0.5572 | 45.32 |
| MGFR(Ours) | 0.2925 | 23.25 | 0.6104 | 0.6854 | 0.8244 | 76.22 |

Table 9: **Quantitative Comparison in CelebA-Test.** Results in red and blue signify the highest and second highest, respectively. The ↓ indicates metrics whereby lower values constitute improved outcomes, with higher values preferred for all other metrics.

## D TRAINING AND INFERENCE CONSUMING ANALYSIS

In terms of training consumption, the proposed MGFR model employs a two-stage training strategy for the dual-control adapter, leading to a moderate increase in training cost. However, for the diffusion-based image restoration model, this additional training time remains relatively short. Nonetheless, this investment is justified, as the proposed MGFR model demonstrates excellent recovery performance. Additionally, the dual-control adapter's specialized design enables superior restoration results depend on the guidance of multimodal information. Our experiments (Figure 32) confirm that employing a single traditional adapter structure for multimodal input often results in redundancy between the

| Method | Real-SR(×8) | | | | | |
| | LPIPS | PSNR | SSIM | ManIQA | ClipIQA | MUSIQ |
|---|---|---|---|---|---|---|
| PSFRGAN | 0.3315 | 22.85 | 0.6232 | 0.6015 | 0.5956 | 73.08 |
| GPEN | 0.3217 | 23.88 | 0.6822 | 0.6754 | 0.6299 | 68.63 |
| VQFR | 0.3277 | 23.16 | 0.6683 | 0.4163 | 0.2363 | 61.92 |
| CodeFormer | 0.3193 | 21.81 | 0.5799 | 0.5970 | 0.6235 | 75.09 |
| DR2 | 0.3580 | 23.26 | 0.6725 | 0.5246 | 0.4494 | 59.46 |
| DiffBIR | 0.3017 | 23.47 | 0.6442 | 0.6058 | 0.7439 | 73.87 |
| BFRffusion | 0.3739 | 23.72 | 0.6718 | 0.4404 | 0.5298 | 42.84 |
| MGFR(Ours) | 0.3227 | 22.34 | 0.5904 | 0.6776 | 0.8083 | 75.94 |

Table 10: **Quantitative Comparison in CelebA-Test.** Results in red and blue signify the highest and second highest, respectively. The ↓ indicates metrics whereby lower values constitute improved outcomes, with higher values preferred for all other metrics.

| Method | Real-SR(×16) | | | | | |
| | LPIPS | PSNR | SSIM | ManIQA | ClipIQA | MUSIQ |
|---|---|---|---|---|---|---|
| PSFRGAN | 0.3788 | 21.27 | 0.5899 | 0.5739 | 0.6274 | 71.76 |
| GPEN | 0.3831 | 22.22 | 0.6541 | 0.6618 | 0.5897 | 66.61 |
| VQFR | 0.3761 | 21.72 | 0.6413 | 0.6513 | 0.2148 | 60.49 |
| CodeFormer | 0.3821 | 21.19 | 0.5717 | 0.5803 | 0.5877 | 70.85 |
| DR2 | 0.3796 | 21.06 | 0.6225 | 0.5160 | 0.5035 | 70.31 |
| DiffBIR | 0.4238 | 21.21 | 0.5654 | 0.5361 | 0.7164 | 67.41 |
| BFRffusion | 0.3735 | 23.67 | 0.6716 | 0.4204 | 0.5098 | 43.16 |
| MGFR(Ours) | 0.3760 | 20.54 | 0.5452 | 0.6729 | 0.7944 | 75.76 |

Table 11: **Quantitative Comparison in CelebA-Test.** Results in red and blue signify the highest and second highest, respectively. The ↓ indicates metrics whereby lower values constitute improved outcomes, with higher values preferred for all other metrics.

reference image and the low-quality input, as well as color inconsistencies in the recovered output. This observation, however, does not preclude further exploration in this area. Our future work will focus on employing a specially designed single-transformer adapter to replace the dual-control adapter, aiming to reduce the model's complexity.

In addition, Table 12 presents the average inference time, memory consumption, parameter count, and FLOPs statistics. Notably, the CFG strategy is compatible with all LDM-based recovery models. Results are presented separately to reflect the CFG strategy's influence during inference. Without the CFG strategy, our model exhibits slightly higher time and memory consumption compared to DiffBIR Lin et al. (2023). DR2 Wang et al. (2023b) and BFRfusion Chen et al. (2024) exhibit faster inference times; however, their recovery performance is suboptimal. Furthermore, SUPIR's large-scale model design results in significantly higher training and testing costs compared to other methods, including MGFR. However, MGFR outperforms SUPIR Yu et al. (2024) on the face image restoration task while incurring lower costs (see Appendix M). It should be noted that efficiency is not the primary focus of this work. Moreover, we believe that the development of efficient lightweight models is grounded in the superior performance of large-scale models. Our future iterations will explore model compression techniques, such as quantization and pruning, to enhance inference speed and reduce parameter counts while maintaining MGFR's superior performance on the face recovery task.

| Method | Average time (s) | Memory consuming (M) | #Params (M) | FLOPs (G) |
|---|---|---|---|---|
| DiffBIR | 5.1 | 11260 | 1716.7 | 897.5 |
| DR2 | 2.6 | 3144 | 93.56 | 388.94 |
| BFRffusion | 3.2 | 8338 | 1197.4 | 784.5 |
| SUPIR | 47.6 | 54318 | 3870.0 | 11950 |
| Ours(w/o CFG) | 6.9 | 15351 | 2029.3 | 890.5 |
| Ours (w/ CFG) | 12.5 | 15351 | 2029.3 | 2672.4 |

Table 12: Inference consuming compared with other diffusion model-based methods.

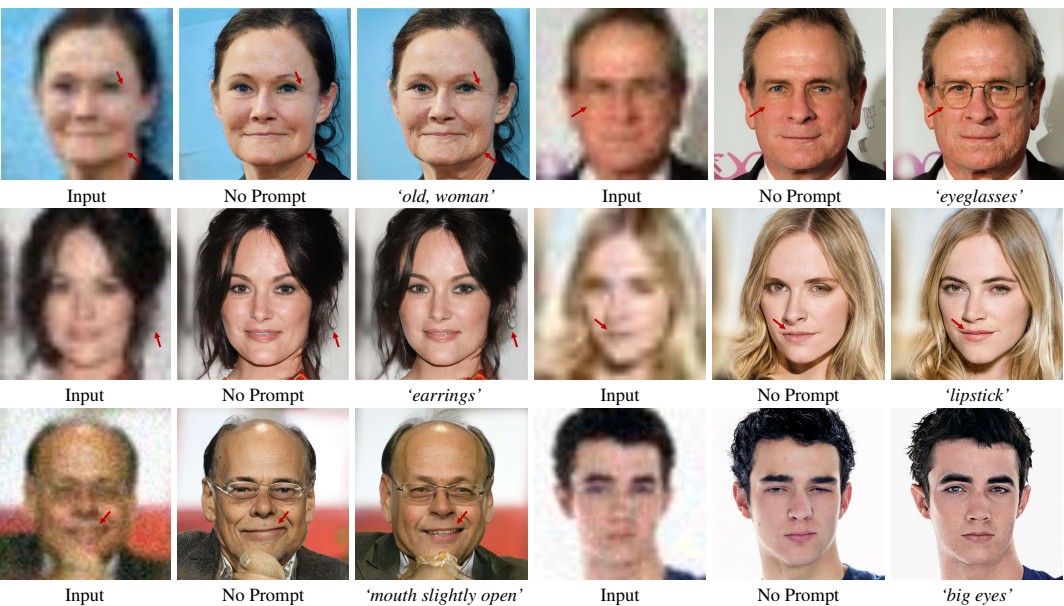

Figure 23: Influences of attribute prompts.

# E CONTROLLING WITH ATTRIBUTES PROMPTS

## E.1 CONTROLLING RESTORATION

Our model facilitates guidance through user-defined attribute prompts during testing. Figure 23 exemplifies this with a demonstration of attribute prompt-controlled recovery. Notably, 'No Prompt' refers to the initial prompt input, 'A high quality, high resolution, realistic, and extremely detailed image.' As illustrates, users can employ prompts like 'old' to define the approximate age in the restored image, or 'eyeglasses' and 'earrings' to add accessories to the image. Furthermore, users can provide additional attribute prompts to refine unsatisfactory results. For instance, 'lipstick' can be used to add lipstick, or 'mouth slightly open' to adjust the mouth's appearance. More significantly, severe illusions, particularly in the eye area, are common in previous methods due to insufficient information in LQ inputs. This observation underscores the importance of attribute prompts in our method, as using 'big eyes' leads to more realistic eye effects. Therefore, we posit that attribute text holds potential as a versatile tool for controlling face recovery.

## E.2 SENSITIVITY ANALYSIS

Moreover, as depicted in Figure 24 case 1 and case 2, with increasing levels of degradation, the model's reliance on attribute prompts for control becomes more apparent, leading to greater flexibility. This observation, a logical experimental outcome, confirms the model's fidelity to LQ inputs during recovery. Specifically, attribute prompts that starkly contradict the LQ input do not influence the effect, which aligns with our expectations. The primary function of attribute labels, we contend, is to facilitate more efficient and effective image restoration, rather than to focus on image editing and control. This is intrinsic to the core objective of real-world face restoration. In our method, all attribute labels listed in Table 7, including 'black hair', 'brown hair', and others, do not possess the ability to control recovery but rather aid the model in interpreting the LQ input. These insights robustly underscore the effectiveness of our approach.

# F USER STUDY

Currently, the relevance and efficacy of metrics such as PSNR, SSIM, and LPIPS require evaluation. In this study, a User study was conducted as an alternative metric for assessing image restoration quality. The study concentrated on two primary questions: (1) How does our model without reference images perform in terms of restoring image quality versus reducing facial illusions compared to

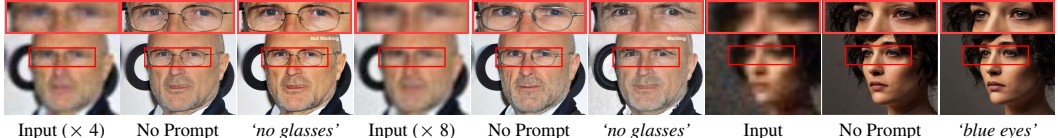

Input ($\times$ 4)    No Prompt    *'no glasses'*    Input ($\times$ 8)    No Prompt    *'no glasses'*    Input    No Prompt    *'blue eyes'*

Figure 24: We investigate the following options for attribute prompt control. First of all, the model becomes increasingly dependent on attribute prompt as input deterioration increases (case 1 & case 2). Second, the input attribute tag does not have a control role if it is not present in Table 7 (case 3).

previous methods? (2) Does the addition of reference image and identity information in guiding restoration result in images that are closer to the Ground Truth compared to the model without reference images? Two sets of questionnaires were prepared, and the study was conducted with 50 participants. Participants were presented with random, anonymous options for their selection. For question (1), our model was compared with DiffBIR Lin et al. (2023), VQFR Gu et al. (2022), and CodeFormer Zhou et al. (2022), focusing on selecting images with better quality and fewer hallucinations, without providing Ground Truth images. This comparison involved 50 sets of images. For question (2), a self-comparison approach was adopted. Specifically, ground truth images were provided, and participants were asked to choose between restoration results with and without reference images, assessing them based on their proximity and realism to the ground truth. In this experiment, 50 pairs of synthetically degraded images were compared.

Subsequently, the first part of the user study, focusing on the improvement of our model in terms of image quality and the reduction of facial illusion, is discussed. The results and detailed information of this study segment are presented in Figure 25 and Figure 26. It was observed that the majority of the 50 participants favored our model for its superior image quality and minimal facial illusions. Reflecting on the recovery results of the advanced method CodeFormer, illustrated in Appendix C, it is noted that while CodeFormer achieves relatively good quality in restored images, considerable facial illusions persist, particularly around the mouth and eyes. In contrast, our method consistently produces high-quality, realistic facial images with minimal facial illusion. These findings underscore the our model's capability to reduce illusion and enhance image quality through negative prompts. Specifically, supported by the diffusion model and LR control adapter, our model is adept at generating realistic high-quality restorations influenced by negative prompts, and it effectively minimizes facial illusions by utilizing an optimal amount of attribute prompts. The synergy of these elements paves the way for further exploration in MGFR.

It is noteworthy that our two-part User study also corresponds to the two-stage development process of the MGFR model. For the second part, the first User study has demonstrated the superior performance of our model, as shown in the figure. Participants generally agreed that adding a guide to the reference image would further achieve superior visual effects.

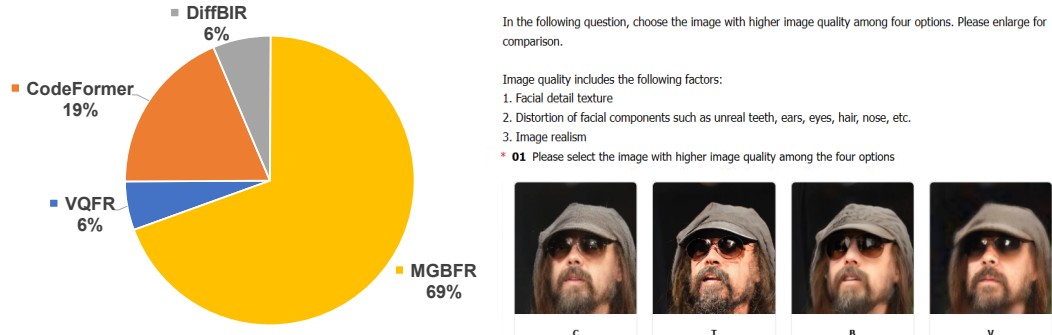

Figure 25: Results and question details of user study.

## G ABLATION STUDY FOR NEGATIVE PROMPT

For negative prompts, we introduce two hyperparameters, $\lambda_{na}$ and $\lambda_{nq}$. However, we find that the changes of the two values tend to have the same effect on the restored images. Thus, we keep

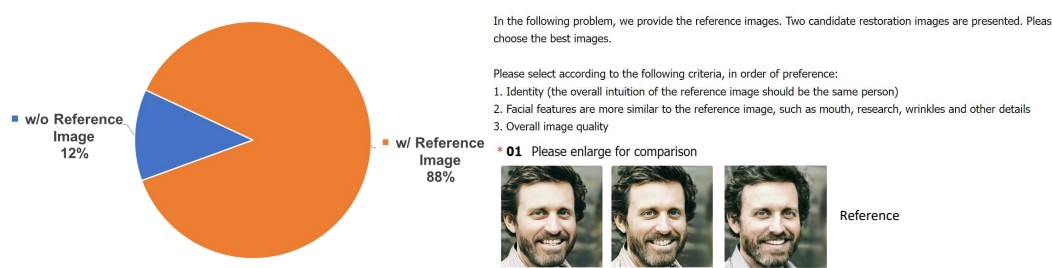

Figure 26: Results and question details of user study.

$\lambda_{na} = 0.5$ and $\lambda_{nq} = 0.5$ during reasoning. Here we will represent the values of $\lambda_{na}$ and $\lambda_{nq}$ with $\lambda$ to show the qualitative comparison results under different hyperparameters in Figure 27.

## H  IMPACT STATEMENTS

Controlled generation technology, as a pivotal innovation in the field of diffusion models, exerts a significant impact across multiple sectors of society. In the creative industries, it enables artists and designers to realize complex visions with unprecedented precision and flexibility, fostering innovation in digital art, design, and multimedia content creation. In commercial applications, controlled generation technology enhances marketing strategies by offering more targeted and dynamic advertising visuals, effectively engaging consumers. Additionally, its influence extends to education and training, where it can revolutionize teaching methods and materials, especially in visually-dependent disciplines, by generating customized educational content and simulations.

The work presented in this paper aims to advance machine learning and computer vision. This method can provide the public with better face processing effects and has greater social value. However, the technique is designed to process facial information, inevitably involving facial attributes such as race and privacy risks. We are aware of these risks. Our research uses publicly available data and images accompanied by captions. We are also wary of potentially discriminatory attribute descriptions in our research. Our method also provides control over face restoration, which reduces the possibility of our method outputting harmful information.

## I  MORE QUALITATIVE COMPARISONS FOR MGFR MODEL

Figure 28 displays the qualitative comparison results between the proposed MGFR model and other advanced methods. The "w/o Reference Image" represents the restoration results of our model after initial training. The use of the negative intuition strategy and attribute prompts significantly reduces the false illusions in face images and substantially enhances overall quality. Subsequently, the inclusion of additional multi-modal information, such as reference images and identity information, can achieve superior visual effects.

## J  SCALABILITY OF MGFR FOR REAL-WORLD VIDEO FACE RESTORATION

The proposed MGFR framework shows significant potential for real-world video-based face recovery tasks. Unlike single-image restoration, video restoration poses the unique challenge of ensuring temporal consistency. To address this, our method leverages the recovered output of the previous frame as a reference for the current frame. This approach aligns seamlessly with our model architecture, which integrates high-quality continuous frame references into guided restoration. Additionally, as our model does not require strict alignment between the reference and low-quality inputs, it effectively handles natural variations in pose and expression commonly found in consecutive video frames, surpassing previous reference-based face restoration models. By leveraging temporal dependencies between frames, the proposed method ensures identity consistency and high-quality recovery in video sequences. Future work could enhance this approach by integrating explicit temporal models or constraints, such as optical flow guidance, to better handle motion artifacts and dynamic variations

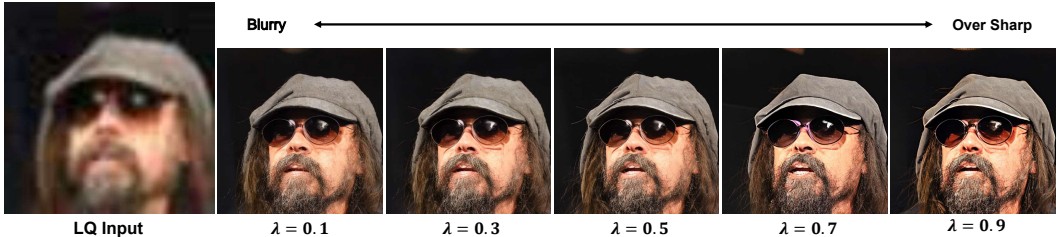

Figure 27: Influence of hyperparameters on recovery effect in CFG. The smaller $\lambda$ does not get a clear recovery result and the huge $\lambda$ causes the recovered image to be over sharp.

in video data. Unlike single-image restoration based on reference images, video data offers more diverse and abundant training samples, which we believe will further unlock the potential of our proposed model. This will be a key focus of our future work.

Figure 28: More qualitative comparisons for MGFR with reference image and ID guidance on synthetic dataset in Reface-Test dataset. Zoom in for best view.

## K    MODEL STABILITY

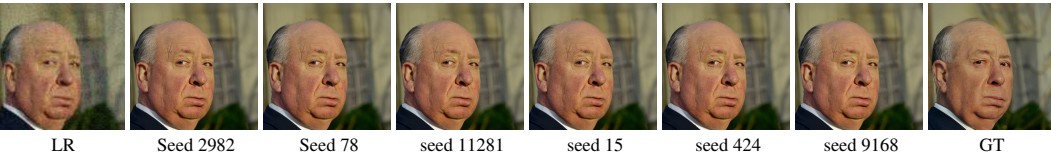

| LR | Seed 2982 | Seed 78 | seed 11281 | seed 15 | seed 424 | seed 9168 | GT |

Figure 29: **Model Stability Analysis**. The recovery results of MGFR remain consistent across different random seeds, eliminating the need for selection among multiple input outcomes.

## L    BRIEF OVERVIEW OF EVALUATION METRICS

For quantitative comparison, the selected image quality evaluation metrics include full-reference metrics PSNR, SSIM, and LPIPS Zhang et al. (2018). Yu et al. (2024); Jinjin et al. (2020) experiment initially confirmed that as image restoration quality improves, the reference utility of metrics such as PSNR, SSIM, and LPIPS needs to be re-evaluated, necessitating the selection of more effective evaluation indicators. Therefore, we introduce three non-reference metrics—ManIQA Yang et al. (2022), ClipIQA Wang et al. (2023a), and MUSIQ Ke et al. (2021)—in this work.

A summary of each evaluation metric is provided below.

- **SSIM** is a key metric for assessing image restoration quality, measuring the similarity between the restored and original images based on brightness, contrast, and structural information. It has been widely used in previous face image restoration tasks Lin et al. (2023); Wang et al. (2023b; 2021a); Yang et al. (2021a); Zhou et al. (2022); Gu et al. (2022); Yu et al. (2024); Chan et al. (2021); Chen et al. (2021); Li et al. (2023; 2020b;a); Wang et al. (2021b); Li et al. (2020c); Teng et al. (2022).

- **PSNR** is a metric derived from the mean square error (MSE), calculated as the logarithmic ratio of the maximum possible pixel value to the error. The results are expressed in decibels (dB), where higher values signify better image quality. It has been widely used in previous face image restoration tasks Lin et al. (2023); Wang et al. (2023b; 2021a); Yang et al. (2021a); Zhou et al. (2022); Gu et al. (2022); Dogan et al. (2019); Yu et al. (2024); Chan et al. (2021); Chen et al. (2021); Li et al. (2023; 2020b;a); Wang et al. (2021b); Li et al. (2020c); Teng et al. (2022).

- **LPIPS** quantifies image differences by extracting features from deep neural networks and measuring the distances between these features. This metric better captures perceptual changes in image details and textures. Previous studies have emphasized image similarity metrics aligned with human visual perception Lin et al. (2023); Wang et al. (2023b; 2021a); Yang et al. (2021a); Zhou et al. (2022); Gu et al. (2022); Yu et al. (2024); Chan et al. (2021); Chen et al. (2021); Li et al. (2023; 2020b;a); Wang et al. (2021b); Li et al. (2020c); Teng et al. (2022).

- **ManIQA** maps images into a low-dimensional manifold space and analyzes their feature distribution and location to assess image quality. This approach demonstrates a high correlation with perceived quality, and its effectiveness has been validated in Yu et al. (2024).

- **MUSIQ** implements a multi-scale feature extraction mechanism designed to capture the quality characteristics of images across varying resolutions and perceptual scales for effective image quality evaluation, and its effectiveness has been validated in Yu et al. (2024).

- **ClipIQA** leverages the robust vision-language priors embedded within the CLIP model. The focus is on enhancing the capability to evaluate both quality perception (seeing) and abstract perception (feeling) of visual content. This approach's effectiveness has been demonstrated in Yu et al. (2024).

## M    QUALITATIVE COMPARISON WITH SUPIR

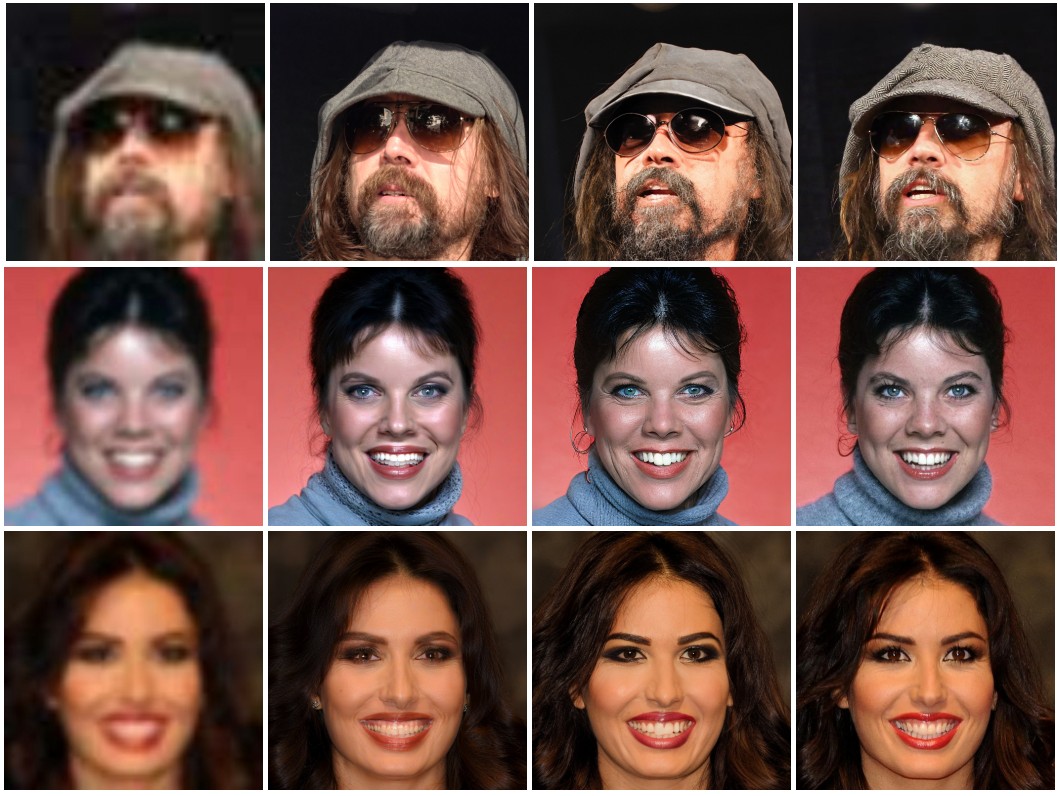

| LR | SUPIR | Ours w/o Ref. | GT |

Figure 30: Qualitative comparisons with SUPIR Yu et al. (2024) for our text-guided baseline model on synthetic dataset under moderate degradation in CelebA-Test dataset. Zoom in for the best view.

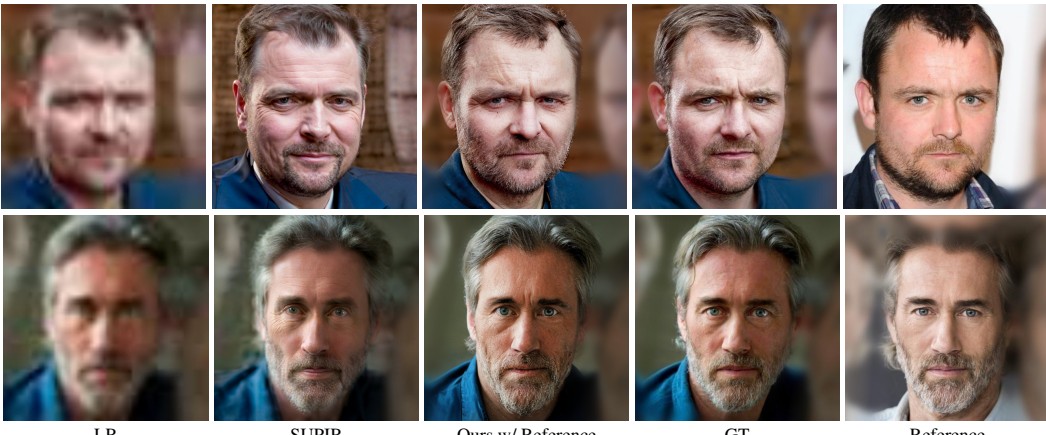

| LR | SUPIR | Ours w/ Reference | GT | Reference |

Figure 31: Qualitative comparisons with SUPIR Yu et al. (2024) for MGFR on synthetic dataset under moderate degradation in Reface-Test dataset. Zoom in for best view.

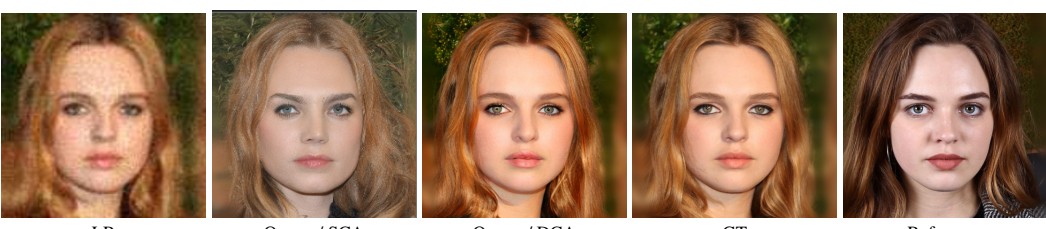

| LR | Ours w/ SCA | Ours w/ DCA | GT | Reference |

Figure 32: Ablation experiments comparing the reception of multi-modal information using a single control adapter (SCA) versus a dual control adapter (DCA) revealed that SCA led to reduced recovery performance and increased chromatic aberration.

