# OpenReview forum: "Overcoming False Illusions in Real-World Face Restoration with Multi-Modal Guided Diffusion Model"
_ICLR.cc/2025/Conference — ICLR 2025 Spotlight_

### Official Review · Reviewer_bNtm · 2024-10-31

**Soundness:** 3
**Presentation:** 3
**Contribution:** 3
**Rating:** 6
**Confidence:** 4

**Summary:**

The paper introduces a Multi-modal Guided Real-World Face Restoration (MGFR) approach, which aims to enhance facial image restoration from low-quality inputs by leveraging multi-modal prior information. This includes attribute text prompts, high-quality reference images, and identity information to mitigate false facial attributes and identities typically generated by current restoration methods. MGFR employs a dual-control adapter and a two-stage training strategy to effectively utilize these priors for targeted restoration tasks. The paper also introduces the Reface-HQ dataset, containing over 23K high-resolution facial images from 5K identities, to support the training of reference-based restoration models. The proposed approach demonstrates superior performance in restoring facial details, particularly under severe degradation, while ensuring identity preservation and attribute correction.

**Strengths:**

* Novel framework:

The MGFR approach introduces a novel dual-control adapter and two-stage training strategy, effectively combining multi-modal priors for enhanced face restoration. The use of attribute text prompts alongside high-quality reference images and identity information is a significant advancement, offering more control over the restoration process.

* Dataset Contribution:
The introduction of the Reface-HQ dataset addresses a critical gap in the availability of high-resolution reference images, providing a valuable resource for the community.

* Good Performance:
The method achieves superior visual quality in restoring facial details under severe degradation, demonstrating its practical applicability in real-world scenarios. The ability to control restoration through textual prompts enhances the flexibility and precision of the restoration process.

* Mitigation of False Illusions: MGFR effectively addresses the problem of false facial attributes and identities, a common issue in existing generative face restoration methods.

**Weaknesses:**

* Complexity of Implementation:
The integration of multiple modalities and the dual-control adapter may introduce complexity in implementation. Therefore, I recommend that this paper should consider a more detailed information on computational requirements and scalability would be beneficial.

* Evaluation Metrics:

While the paper demonstrates superior performance, a more thorough discussion of the evaluation metrics used to assess visual quality, identity preservation, and attribute correction would strengthen the claims.

* Ablation Studies:
The paper would benefit from additional ablation studies to isolate the impact of each component (e.g., attribute prompts, reference images) on the overall performance, providing deeper insights into the effectiveness of each modality.

**Questions:**

1) How will the performance vary if different versions of diffusion models are used? That is, would diffusion models be the strong foundation of achieving the good performance?

2) Some figures, say fig.4 and 5, are with too small fonts and unclear color denotions, making it less readable.

**Details Of Ethics Concerns:**

human face images are collected for training the proposed model.

---

> ### Author Response · Authors · 2024-11-22
> **Response to Reviewer bNtm (1/2)**
>
> We sincerely appreciate the reviewer’s recognition of our work, particularly the positive feedback on the novel framework of the MGFR approach, the contribution of the Reface-HQ dataset, and the superior performance of our method. We are also grateful for the reviewer’s acknowledgment of our efforts in integrating multi-modal priors, addressing real-world degradation, and mitigating the generation of false facial attributes.
>
> `1.` *Complexity of Implementation*
>
> `A`: We thank the reviewer for the insightful comments. We acknowledge that the inclusion of additional functionalities in the proposed MGFR model may introduce higher implementation complexity and computational demands. However, for diffusion-based methods like ours, the diffusion model itself is inherently computationally intensive. Recent state-of-the-art models such as SUPIR and DiffBIR, which are also diffusion-based, require substantial computational resources and extensive training.
>
> In response to the reviewer’s suggestion, we have included a more detailed discussion in the revised manuscript, comparing the computational requirements (e.g., FLOPs and parameter counts) of our model with those of other contemporary methods. The results indicate that our approach is not significantly more complex than similar methods. However, in the context of face-focused restoration tasks, our method demonstrates significantly better performance compared to competing approaches.
>
> Additionally, we recognize the potential for further optimizations to make our method more practical for real-world applications. Techniques such as low-bit quantization for inference, model pruning, and distillation can substantially enhance efficiency. While these topics are beyond the scope of this paper for an in-depth exploration, we have added relevant discussions in the supplementary material and plan to address these optimizations in future work.
>
> `2.` *Evaluation Metrics*
>
> `A`: Thank you for your question. Assessing the perceptual quality of images is indeed a challenging task, as extensively discussed in prior works. It is difficult for any single metric to accurately represent objective quality. In our paper, we include six different objective image evaluation metrics: SSIM, PSNR, LPIPS, ManIQA, ClipIQA, and MUSIQ, which encompass traditional image similarity metrics, deep learning-based metrics, and unsupervised metrics that are more suitable for generative models. These metrics are widely used in the field of perceptual image processing.
> Additionally, we include the ArcFace identity distance metric, which is extensively applied in works related to face recognition and face generation. All objective metrics demonstrate the effectiveness of our method. Furthermore, we conducted a user study to provide additional evidence of the superiority of our proposed approach. The results and analysis from this study have been included in the updated manuscript.
>
> `3.` *Ablation Studies*
>
> `A`: Thank you for your valuable suggestions. In this paper, we conducted the following ablation experiments:
>
> 1. *Model performance without reference images (Table 1)*: The model achieves state-of-the-art performance and outstanding restoration quality even without reference image input.
> 2. *Model performance with reference images (Table 2)*: The model achieves better restoration results and lower identity information loss when guided by reference images.
> 3. *Controlled restoration using facial attribute prompts (Figure 8)*: MGFR effectively uses attribute prompts to edit and control the restoration results.
> 4. *Ablation study on the effect of reference images (Figure 11)*: Demonstrates the effectiveness of reference images in guiding restoration results.
> 5. *Ablation study on identity information integration (Figure 15)*: Incorporating ArcFace identity embeddings significantly reduces identity information loss.
> 6. *Ablation study on negative prompts (Figure 13, Table 4)*: Negative prompts substantially improve restoration quality and detail reconstruction.
> 7. *Ablation study on additional information exchange (Table 3)*: Incorporating additional information exchange enhances the control effect of reference images.
> 8. *Experiments on attribute prompts completely inconsistent with low-quality inputs (Figure 10)*: Shows that the model consistently adheres to low-quality inputs during restoration.
> 9. *Experiments on the sensitivity of attribute text control (Appendix E.2)*: Validates that the sensitivity of attribute text control is positively correlated with the degree of degradation, as expected.
>
> If the reviewer believes additional ablation studies are necessary, we are more than happy to discuss and conduct further tests if needed.

---

> ### Author Response · Authors · 2024-11-22
> **Response to Reviewer bNtm (2/2)**
>
> `4.` *How will the performance vary if different versions of diffusion models are used? That is, would diffusion models be the strong foundation of achieving the good performance?*
>
> `A`: We appreciate your insightful feedback. More advanced diffusion models, such as SDXL used in the SUPIR model and the newer SD3 model, can indeed be applied to our approach and achieved promising results. This is primarily due to the heavy reliance of generative model training on extensive image datasets. Without leveraging pre-trained generative models, we would require significantly more data to achieve comparable performance and intelligence. Many state-of-the-art intelligent image processing diffusion models, such as SUPIR, DiffBIR, PASD, and SeeSR, are based on pre-trained diffusion models.
>
> **However, utilizing pre-trained models is only part of what makes our approach successful.** The exceptional restoration results of our method are primarily attributed to our unique model design, architecture, and the meticulous choices in training strategies and data. It is worth noting that while adopting more advanced diffusion models could further enhance performance, these models typically come with higher training complexity and resource requirements. This consideration motivates us to explore incorporating advanced diffusion models into future iterations of MGFR.
>
>
> `5.` *Some figures, say fig.4 and 5, are with too small fonts and unclear color denotions, making it less readable.*
>
> `A`: Thank you for your valuable feedback. In the revised manuscript, we have carefully adjusted the font sizes and included clearer color annotations to enhance readability and clarity.

---

### Official Review · Reviewer_2kxB · 2024-11-02

**Soundness:** 4
**Presentation:** 3
**Contribution:** 3
**Rating:** 8
**Confidence:** 5

**Summary:**

This work addresses the challenge of real-world face restoration. The authors introduce the Multi-modal Guided Real-World Face Restoration (MGFR) method, which improves facial image restoration from low-quality inputs using attribute prompts and reference images. Specifically, MGFR employs a dual-control adapter and a two-stage training strategy, leveraging a dataset of over 23,000 high-resolution images to enhance visual quality, identity preservation, and attribute correction. Experimental results demonstrate that the proposed method achieves superior performance.

**Strengths:**

1. The topic of real-world face restoration is interesting.
2. The overall writing of the manuscript is clear and easy to follow.
3. This work introduces the Reface-HQ dataset, which comprises over 23,000 high-resolution images. This dataset can provide a foundation for training and evaluating the model.

**Weaknesses:**

1. The proposed method appears complex, and its running time and memory consumption are not superior to existing methods like DiffBIR and DR2. Since running time and memory usage are crucial for real-world applications, this raises concerns about its practical feasibility.
2. There is a lack of comparison in FLOPs and Params in the main comparison and ablation study.
3. Lacking comparison to recent works (SUPIR, BFRffusion) in Table 1.
a. Scaling up to excellence: Practicing model scaling for photo-realistic image restoration in the wild
b. Towards real-world blind face restoration with generative diffusion prior.

**Questions:**

1. Not super important, but how do the authors feel the proposed method can be extended for real-world video face restoration? Can they please add some discussion on the scalability of the proposed method?
2. Overall the paper looks promising and makes meaningful contributions, however, it lacks some important experiments and details. The authors can refer to the weaknesses section.

---

> ### Author Response · Authors · 2024-11-23
> **Response to Reviewer 2kxB**
>
> We thank the reviewer for recognizing our work, the MGFR method, and the Reface-HQ dataset, as well as for the thoughtful feedback.
>
> `1.` *The proposed method appears complex*
>
> `A`: We appreciate the reviewer’s insightful comments. While we recognize that the additional functionalities in the proposed MGFR model may increase implementation complexity and computational demands, it is important to note that diffusion-based methods inherently involve high computational costs. Recent state-of-the-art models, such as SUPIR and DiffBIR, which are also diffusion-based, similarly require substantial computational resources and extensive training. To address the reviewer’s concerns, we have expanded the discussion in the revised manuscript to include a detailed comparison of computational requirements, such as FLOPs and parameter counts, between our model and other contemporary methods. The results indicate that our approach is not significantly more complex than similar methods, yet it demonstrates notably superior performance for face restoration tasks. Additionally, we acknowledge the potential for further optimizations, including low-bit quantization, model pruning, and distillation, to enhance efficiency and make our method more practical for real-world applications. While these optimizations are beyond the scope of this paper, we have included a discussion of these possibilities in the supplementary material and plan to explore them in future work.
>
> `2.` *There is a lack of comparison in FLOPs and Params in the main comparison and ablation study.*
>
> `A`: Thank you for your question. In the revised manuscript, we include detailed information on FLOPs and the number of parameters in the Table 12, along with a comparison to SUPIR. It is important to note that the FLOPs of the diffusion model depend on the number of inference steps, which can vary. To provide consistency, we will report results based on a single inference step.
>
> `3.` *Lacking comparison to recent works (SUPIR, BFRffusion)*
>
> `A`: Thank you for your question. We have conducted supplementary experiments using the same low-quality inputs as in the original study, and the quantitative comparison results with BFRFusion are now included in Tables I, 9, 10, and 11 of the revised manuscript. These results demonstrate that our model continues to achieve state-of-the-art performance.
>
> Regarding SUPIR, its batch inference requires substantial hardware resources and computation time, making it challenging to complete all quantitative evaluations within the short discussion phase. We will continue these tests and include the results in future updates of the manuscript once completed. In the meantime, we have provided qualitative comparison results with the SUPIR model in Appendix K of the revised manuscript. These results show that while SUPIR can achieve high-quality outputs, it often loses significant identity information. In contrast, MGFR effectively preserves this information through the incorporation of reference images, which highlights the key contribution and significance of our approach.
>
> `4.` *how do the authors feel the proposed method can be extended for real-world video face restoration?*
>
> `A`: Thank you for the reviewer’s question. We are currently conducting research on video restoration based on diffusion generative models. The primary challenge in extending single-frame methods to video lies in ensuring inter-frame consistency. Our method, with its ability to inherit key features and identity information from reference images, effectively addresses semantic-level consistency, making it better suited for video restoration compared to other single-frame models. However, our approach still needs to overcome pixel-level consistency issues to prevent flickering between frames. This requires pixel-level referencing and smoothing, as well as handling temporal information effectively. Future improvements may involve incorporating temporal inputs and modifying the decoder to output smoother multi-frame results. Nevertheless, we are confident that using relevant frames as reference inputs, which is a core aspect of our method, is one of the key factors for high-quality video restoration.
>
> `5.` *Overall the paper looks promising and makes meaningful contributions, however, it lacks some important experiments and details.*
>
> `A`: We thank Reviewer 2kxB for recognizing the meaningful contribution of our paper and its potential, as highlighted in your comments. We have supplemented and revised the questions that have been raised, improved the relevant experiments, and included additional experimental data and detailed discussions in the revised manuscript. Should there be any further questions or suggestions, we would be more than happy to engage in further discussions. Once again, we deeply value your review and your recognition of our efforts.

---

> > ### Comment · Reviewer_2kxB · 2024-11-27
> >
> > Thank you for your response.  I have carefully checked the authors' response.  The overall writing of the paper is good and easy to follow.  In addition, this work introduces the Reface-HQ dataset, which is benefit for the community.  I raise my scroe.

---

> > > ### Author Response · Authors · 2024-11-27
> > >
> > > We thank the reviewers for their valuable feedback and thoughtful discussions, as well as their recognition of our work. We will incorporate the reviewers' suggestions to further enhance the paper.

---

### Official Review · Reviewer_1C8W · 2024-11-04

**Soundness:** 3
**Presentation:** 3
**Contribution:** 2
**Rating:** 8
**Confidence:** 4

**Summary:**

The paper propose to utilize diffusion prior, as long as multi-modality input to perform real world face restoration. The method can receive text or reference image for face restoration.

They also present a dataset that contains identity-image pair.

There is both quantitative  and qualitative metric, which demonstrate the effectiveness of this method.

**Strengths:**

1. The motivation that using multi-modality input to assist face restoration is nice. As shown in the paper, there could be ambiguity during restoration, and the additional input could be helpful.
2 .  The use of diffusion prior for this work also make sense, and it models a probability given the input.
3. The dataset could be beneficial to the community.

**Weaknesses:**

1. The method used in the paper is not new. The main contribution of this works seems to be on the dataset and using expositing method on a newer task.
2. Ablation of the effectiveness of using  trained diffusion prior is missing.

**Questions:**

1. The design of the Dual control Adapter is a little but tricky. Can one just replace it by a transformer?
    2. There can be different  result for every sample. How diverse is the model? And how to decide which sample to use?
    3. Will be dataset be realised?

---

> ### Author Response · Authors · 2024-11-23
> **Response to Reviewer 1C8W**
>
> We thank the reviewer for recognizing our work, particularly the motivation for using multi-modal inputs to assist face restoration, the rationale of diffusion prior, and the potential contribution of the Reface-HQ dataset to the community.
>
> `1.` *The method used in the paper is not new.*
>
> `A`: We respectfully disagree with the reviewer’s comment. Our method is not a simple application of existing techniques. Specifically:
>
> 1. We are the first to propose to simultaneously use attribute prompts, high-quality reference images, and face identity information in a generative model to guide real-world human image restoration.
> 2. Based on face image datasets, we extracted facial attributes and integrated them as attribute prompts through LLMs, enabling prompt-based editing and controlled restoration.
> 3. We designed a counterintuitive strategy that leverages negative prompts to further enhance restoration quality and facial details.
> 4. MGFR improves restoration results and reduces facial identity loss under the guidance of reference images, without requiring alignment between the reference and low-quality input images.
> 5. To effectively analyze and integrate multi-modal inputs, we introduced a dual-control adapter and a two-stage training strategy.
>
> We believe our work demonstrates significant innovation.
>
> `2.` *Ablation of the effectiveness of using trained diffusion prior is missing.*
>
> `A`: Thanks for the feedback. Recent advanced face restoration methods based on diffusion models (e.g., DiffBIR, SUPIR, and BFRFusion) leverage pre-trained diffusion priors to achieve robust performance. These pre-trained diffusion priors significantly reduce training difficulty and time, allowing models to better learn data distributions consistent with specific tasks, thereby enhancing their effectiveness. Consequently, the use of pre-trained diffusion priors has become a common practice in diffusion-based face restoration models.
>
> Ablation studies on the effectiveness of pre-trained diffusion priors would require training models from scratch, demanding extensive time, computational resources, and data, making it challenging to complete within a limited timeframe. Furthermore, numerous existing works have demonstrated that training diffusion models from scratch with limited data often fails to produce satisfactory results.
>
> That said, utilizing pre-trained models is only one aspect of our method's success. The exceptional restoration results achieved by our approach are primarily attributed to our unique model design, innovative architecture, and the careful selection of training strategies and data. While incorporating more advanced diffusion models could potentially improve performance, such models generally come with increased training complexity and resource demands. This drives us to consider integrating advanced diffusion models in future iterations of MGFR to further refine our approach.
>
> `3.` *The design of the Dual control Adapter is a little but tricky. Can one just replace it by a transformer?*
>
> `A`: Thank you for your question. We also aimed to achieve the desired functionality with a single control module. However, our experiments showed that using a single controller to simultaneously handle guidance information from multiple modalities led to issues such as color discrepancies in the restored output, loss of control, and even suboptimal results. To successfully accomplish the task of controlling face image restoration using facial identity information and reference image guidance, we carefully designed a dual-controller scheme. Designing a single controller would require further investigation into the source of the control effect in diffusion models. We look forward to exploring whether a specially designed single-transformer architecture could further optimize our approach in future work.
>
> `4.` *There can be different result for every sample. How diverse is the model? And how to decide which sample to use?*
>
> `A`: Thank you for raising this insightful question. To evaluate the diversity and stability of our model, we conducted experiments by performing inference under multiple random seeds. The results demonstrated minimal variation across different runs, indicating that the model produces consistent outputs with negligible randomness. This stability ensures the practical usability of our method, as users can reliably achieve high-quality results without significant deviations.
>
> `5.` *Will be dataset be realised?*
>
> `A`: Thank you for your question. The proposed Reface-HQ dataset is set to be released. Compared to existing datasets, Reface-HQ offers a greater number of identity face images along with increased training and testing samples. In addition to the 512-resolution dataset utilized in this work, a 1024-resolution subset will also be made available for training and evaluating higher-resolution models. This dataset enables previous models to be leveraged to their full potential.

---

> > ### Comment · Reviewer_1C8W · 2024-11-26
> >
> > Thanks for the response.
> >
> > For me, the neg-prompts inference, multi-modal conditioned diffusion network is still not new. But the authors do a good job in building a system that works in face restoration using those techniques. I also acknowledge the motivation of using multi-modal information to assist face restoration, a task that could be ambiguous.
> >
> > Also considering the fact that the dataset will be released and benefit the community, I will raise my score.

---

> > > ### Author Response · Authors · 2024-11-27
> > >
> > > We sincerely thank the reviewers for their recognition of our work and their responses and valuable discussions. We will further improve the paper based on the reviewers' comments. We also hope that the Reface-HQ data can contribute to the community in the future.

---

### Author Response · Authors · 2024-11-26

We sincerely thank all reviewers and area chairs for their valuable comments and recognition of our work, and we are pleased that all reviewers have responded positively to our paper, particularly their acknowledgment of the novel MGFR framework, the contribution of the Reface-HQ dataset, and the method’s outstanding performance. We also appreciate the reviewers’ recognition of our efforts in integrating multimodal priors, addressing real-world degradation, and reducing the generation of false facial attributes.

Reviewers 1C8W (reviewer 1), 2kxB (reviewer 2) and bNtm (reviewer 3) provided insightful comments. In response to their feedback, we have provided detailed clarifications, conducted further analyses, and included additional experiments and data where necessary. Our response is summarized as follows:

1. In response to the concerns raised by reviewers 1, 2, and 3 regarding the complexity of dual-control adapters in the model and the feasibility of applying MGFR’s inference consumption in real-world scenarios, we added a discussion in Appendix D on the necessity of dual-control adapters and identified simplifying their design as a key direction for future work. Table 12 was updated to include the FLOPs and total model parameters for each diffusion-based face restoration method during inference. When compared to state-of-the-art methods, our model demonstrates comparable resource consumption. Finally, we believe that the development of efficient lightweight models is grounded in the superior performance of large-scale models, and future work will incorporate model compression techniques, including quantization and pruning, into MGFR.

2. In response to reviewer 1’s question regarding the selection of samples for MGFR after repeated inference, we included an evaluation of model stability in the revised manuscript, detailed in Appendix K, demonstrating that recovery results remain consistent across different random seeds.

3. In response to reviewer 2’s mention of recent face restoration models (BFRffusion and SUPIR), we incorporated a quantitative comparison of BFRffusion into the revised manuscript and, given time constraints, included a qualitative comparison of SUPIR in Appendix M. The results demonstrate that the proposed MGFR achieves superior recovery performance, particularly in preserving face identity information.

4. In response to reviewer 3’s concerns regarding evaluation metrics and suggestions for improving Figures 4 and 5, we have provided a summary of the evaluation indicators in Appendix L of the revised manuscript and enhanced Figures 4 and 5 to improve their readability.

5. Motivated by reviewer 2’s suggestion to extend MGFR to video face restoration, we included a detailed discussion of this application in the revised manuscript and identified it as a key direction for future research, as detailed in Appendix J.

In conclusion, we are the first attempt to use multimodal information to guide real world face image restoration. We are the first reference image-based face image restoration method that does not require strict face alignment. Qualitative and quantitative evaluations demonstrate that the proposed MGFR model delivers state-of-the-art performance. Future efforts will focus on developing lightweight model variants and streamlined designs, while MGFR has already established a new baseline for real-world face restoration tasks.

Finally, thanks again to all reviewers for their positive feedback and recognition of our work. In the new version of the manuscript, the revisions are marked in blue. If there are any other questions, we would be more than happy to engage in further discussions.

The best,

Authors

---

### Meta-Review · Area_Chair_RErD · 2024-12-15

**Metareview:**

The paper presents the Multi-modal guided face restoration method, which significantly improves face restoration from low-quality inputs using attribute prompts and reference images. Using the Reface-HQ dataset of over 23,000 images, it enhances visual quality and identity accuracy, demonstrating good performance through rigorous experiments.

***Strengths:***
-  Propose a novel dual-control adapter for facial image restoration.
-  Introduce a large-scale, diverse dataset to train and validate the model. The authors promise to release this one for community.
-  Outperform existing methods in visual quality and identity preservation.

***Weaknesses:***
The model's complexity is a concern, although addressed in the authors' rebuttal.
More evidence of generalization  performance on real cases  would strengthen the claims.

After careful consideration and discussion, we are pleased to accept this submission. The decision to accept this paper is based on the innovative approach, strong experimental results, and positive reviewer feedback. The authors have partially addressed the concerns about model complexity, enhancing the paper's contribution to the field of face image restoration.

**Additional Comments On Reviewer Discussion:**

All the concerns have been addressed and the reviewers raised score.

---

### Decision · Program_Chairs · 2025-01-22

Accept (Spotlight)